# Short-range template switching in great ape genomes explored using pair hidden Markov models

**Conor R. Walker**[1,2], **Aylwyn Scally**[2], **Nicola De Maio**[1], **Nick Goldman**[1] *

**1** European Molecular Biology Laboratory, European Bioinformatics Institute (EMBL-EBI), Wellcome Genome Campus, Hinxton, United Kingdom, **2** Department of Genetics, University of Cambridge, Cambridge, United Kingdom

* goldman@ebi.ac.uk

**Data Availability Statement:** All code files are available from https://github.com/conorwalker/tsa_pairhmm and https://github.com/conorwalker/template_switching All relevant data are within the manuscript and its Supporting information files.

## Abstract

Many complex genomic rearrangements arise through template switch errors, which occur in DNA replication when there is a transient polymerase switch to an alternate template nearby in three-dimensional space. While typically investigated at kilobase-to-megabase scales, the genomic and evolutionary consequences of this mutational process are not well characterised at smaller scales, where they are often interpreted as clusters of independent substitutions, insertions and deletions. Here we present an improved statistical approach using pair hidden Markov models, and use it to detect and describe short-range template switches underlying clusters of mutations in the multi-way alignment of hominid genomes. Using robust statistics derived from evolutionary genomic simulations, we show that template switch events have been widespread in the evolution of the great apes' genomes and provide a parsimonious explanation for the presence of many complex mutation clusters in their phylogenetic context. Larger-scale mechanisms of genome rearrangement are typically associated with structural features around breakpoints, and accordingly we show that atypical patterns of secondary structure formation and DNA bending are present at the initial template switch loci. Our methods improve on previous non-probabilistic approaches for computational detection of template switch mutations, allowing the statistical significance of events to be assessed. By specifying realistic evolutionary parameters based on the genomes and taxa involved, our methods can be readily adapted to other intra- or inter-species comparisons.

## Author summary

DNA replication is an imperfect process which causes the mutations that give rise to genetic diversity during the evolution of genomes. While many mutations are independent, single-nucleotide substitutions or small insertions and deletions, some mutations arise as nonindependent clusters of substitutions and larger scale chromosomal rearrangements. Large-scale rearrangements (also called structural variants) in particular can have a profound impact on genome evolution and contribute to both germline and somatic

**Funding:** C.R.W is funded by the National Institute of Health Research (NIHR) Cambridge Biomedical Research Centre; grant number IS-BRC-1215-20014; website: https://cambridgebrc.nihr.ac.uk. C.R.W, N.D.M, and N.G are funded by the European Molecular Biology Laboratory; a specific grant number is not applicable; website: https://www.embl.de. A.S. is funded by the University of Cambridge; a specific grant number is not applicable; website: https://www.cam.ac.uk/. The funders had no role in study design, data collection and analysis, decision to publish, or preparation of the manuscript.

**Competing interests:** The authors have declared that no competing interests exist.

disease in humans. The replication-based mechanisms underlying structural variation typically involve a polymerase switch event in which a large segment of DNA is copied using a template from an alternate location in the genome. Methods for identifying these template switch mutations lack the power to detect smaller scale rearrangements which can arise through the same replication-based pathways. Here we outline a model which can detect and assess the statistical significance of such small-scale template switches within their evolutionary context. We show that these events are widespread in the evolution of great apes and that the genomic features associated with these small-scale rearrangements are similar to those of large-scale structural variants.

## Introduction

Mutation clusters consisting of multiple nearby substitutions and indels (insertions and deletions) in sequence alignments are pervasive throughout eukaryotic genomes [1]. These complex mutation patterns might arise through either a process of random, independent mutation accumulation within a small sequence window, or single mutational events capable of generating many apparent substitutions and indels in a single pass. Other than in small genomic regions that exhibit species-specific accelerated evolution [2], single mutational events provide the most parsimonious explanation for the presence of a complex mutation cluster between two species. Incorrect inference of the evolutionary history of such clusters can have important implications in studies of molecular evolution. For example, methods for inferring adaptive evolution such as the widely used branch-site test rely on likelihood ratio testing, for which a core assumption is that substitutions occur independently and at single sites [3, 4]. When these assumptions are violated, and the branch-site test is applied to regions subject to multi-nucleotide mutations, false inferences of positive selection are produced [5].

In humans, small-scale clustered mutagenesis is typically attributed to local mechanisms that occur during DNA replication, such as error-prone translesion synthesis and replication slippage [6], both of which operate on the nascent strand. Meanwhile, larger scale germline and somatic mutational mechanisms, which can generate kilobase to megabase scale rearrangements through several pathways [7–9], all arise through some form of template switch to an alternate strand. While a process of template switching can occur locally as a mechanism to bypass DNA lesions during replication, it is traditionally considered an error-free pathway mediated by proliferating cell nuclear antigen polyubiquitination, in which replication proceeds in the same direction as the nascent strand following the formation of a hemicatenane structure with the newly synthesised sister chromatid [10–12]. However, error-prone reverse-oriented template switching has now been observed in multiple eukaryotes including humans [13, 14], and has been shown to leave a footprint of clustered mutagenesis in the human genome [15]. Despite our understanding of these individual mechanisms, computationally capturing their mutational footprints in an evolutionary context remains difficult, especially when focusing on local mutational mechanisms, which may present as a plausible cluster of accumulated substitutions and indels within a sequence alignment. The extent to which these processes have shaped human genome evolution is therefore poorly characterised, and a general model which can capture the consequences of any such event is desirable for understanding their role in shaping genome evolution.

The template switch process inherent to all of these replication-based rearrangements involves the dissociation of the 3′ end of the nascent DNA strand and invasion of a physically-close alternate template. A period of replication using this alternate template is then followed

by a second switch event in which the 3′ end of the nascent strand reassociates with the original strand [16], a series of successive switch events that can generate large-scale complex rearrangements [17], or extension of the alternate template until a new telomere is formed [18]. While all these mechanisms require a physically proximal alternate template, there is no requirement that the two regions are nearby in linear sequence space and the position of strand invasion is often mediated solely by small stretches of identity between any two genomic positions regardless of proximity [19–21]. However, attributing a small number of mutations to a short alternate template from any position in a large genome is an intractable problem, and candidate templates with high identity to the focal mutation cluster may readily be found by chance.

Instead, a subset of clustered mutations possibly generated through template switching can be modelled by restricting the search space of potential alternate template to regions in the vicinity (tens to hundreds of nucleotides) of each mutation cluster. Such local (or "short-range") template switching events have been observed *in vivo* during both eukaroytic and prokaryotic replication [22, 23], typically characterised by the conversion of a pre-existing near-perfect inverted repeat sequence into a perfect inverted repeat. To identify such local events in the human genome, Löytynoja and Goldman [15] outlined a mechanism-agnostic "four-point" model for describing short-range template switch events, leveraging a modified dynamic programming approach to parsimoniously explain mutation clusters between closely related species.

The four-point model (Fig 1) assumes switch events occur locally within a single replication fork and captures the consequences of both intra-strand (Fig 1A, left) and inter-strand (Fig 1A, right) switch events, which appear as complex mutation clusters when comparing post-event descendant sequence to pre-event ancestral sequence (Fig 1B). There are no implicit assumptions made about the strandedness of events (the inter-strand switch in Fig 1A is depicted as a leading to lagging strand switch for simplicity), allowing the detection of switch events from either strand. Each event is described using four numbered points, assuming left (Ⓛ) to right (Ⓡ) oriented replication. Points ① and ② describe the genome coordinates of the initial switch event, with dissociation from the nascent strand at ① and strand invasion followed by alternate-template replication at ②. After this transient period of Ⓡ → Ⓛ-orientated replication from ② → ③, a second switch event occurs, with dissociation at ③ and reassociation on the original strand at ④, after which replication proceeds as normal. This four-point notation provides a convenient way to represent the consequences of any single template switch event within an alignment, regardless of the causative mechanism, enabling the definition of three ordered pairwise alignment fragments, Ⓛ → ①, ② → ③ and ④ → Ⓡ, which fully describe any template switch event (Fig 1B).

The consequences of any such template switch process will present as a mutation cluster in a typical pairwise alignment between two closely related sequences (Fig 1B, top), as standard alignment models assume that sequences evolve under single base substitutions and short indels and a combination of these processes is the only way in which the consequences of template switch events can be encoded. In contrast, a template switch alignment aims to model sequence evolution according to both substitutions and indels, as well as an additional single template switch event (Fig 1B, bottom). Assuming a template switch gave rise to an apparent mutation cluster, the template switch alignment of this region will contain appreciably fewer substitutions and indels than the corresponding linear alignment. To determine whether an evolutionary history involving a single template switch is significantly more parsimonious than a combination of single base substitutions and/or indels, it is necessary to compare these two alignment models. This model comparison is not possible under the simple scoring scheme implemented by Löytynoja and Goldman [15], and the statistical significance of any

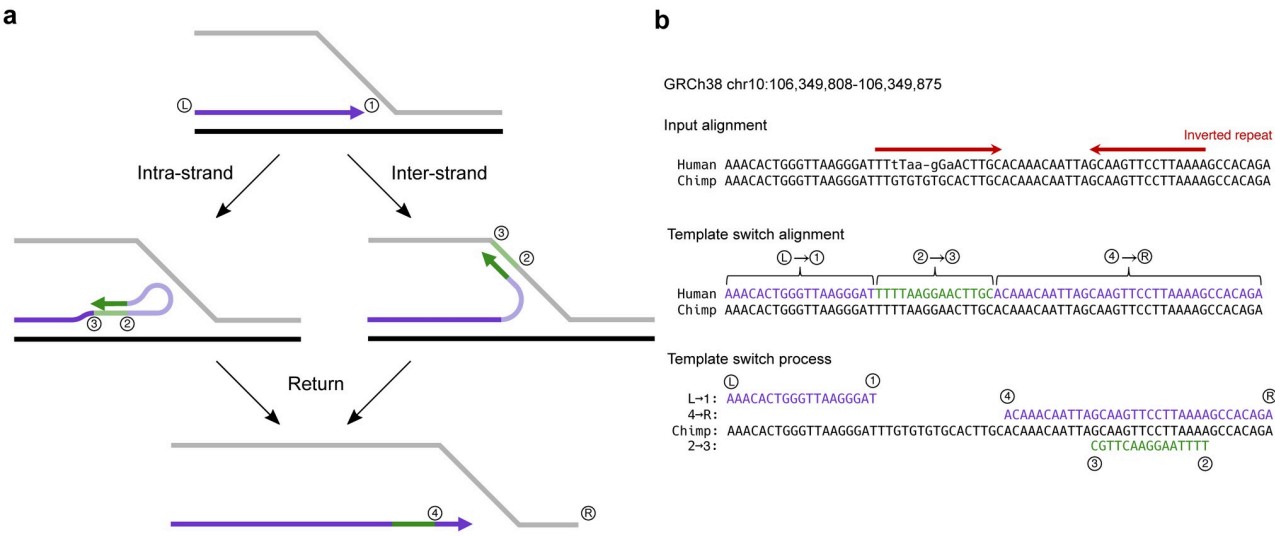

**Fig 1. Diagrammatic representation of a short-range template switch and an example alignment under the four-point model of template switching. (A)** The template switch process projected onto a replication fork. DNA replication (arrow head) is shown proceeding in Ⓛ → Ⓡ orientation (Ⓛ and Ⓡ indicating the assumed direction of replication, not precise locations). A template switch event is initiated at ①; the DNA polymerase dissociates from the nascent strand and attaches at ② (left: intra-strand; right: inter-strand), and replication transiently proceeds in reverse orientation until ③. A second switch event occurs at ③, with the polymerase now detaching from the alternate template region (green lines) and reattaching at ④, from where replication proceeds as normal. This process generates three annotated fragments: the initial and final purple fragments represent the standard-replicated regions, and the central green fragment represents the reverse-replicated region from an alternate template. **(B)** An input alignment between an ancestral and descendant sequence can be scanned to identify a template switch process. In this case, a mutation cluster apparently containing five substitutions and one insertion (top, lower case and − characters) is observed in the alignment between region chr1:36,857,456-36,857,523 of the reference human genome and the chimpanzee genome (Ensembl v.98, EPO alignments of thirteen primates [24]). Under a model of template switching as described above, this mutation cluster can instead be explained with 100% identity by three ordered alignment fragments (middle; S1 Data, event 28). The sequence representation of the template switch process that generates the three alignment fragments is also shown (bottom), with purple and green sequences representing the descendant fragments and the black sequence representing the original, non-mutated strand. Note that the reverse-oriented replication that generates fragment ② → ③ manifests as reverse complement sequence in the descendant with respect to the ancestral template, often generating perfect inverted repeats in the descendant sequence (red arrows above the EPO alignment).

particular event cannot be established. In addition, Löytynoja and Goldman left unresolved issues regarding underestimation of template switch event prevalence and the unknown evolutionary direction of individual events [15].

We therefore introduce a probabilistic method for modelling template switch mutations, allowing us to assign statistical significance to candidate events, and use this model to investigate events across the great ape genomes in their phylogenetic context. We achieve greater resolution in the detection of short-range template switch events across the human reference genome and identify thousands of significant events across the great ape tree. We present distinct physical properties of the DNA duplex surrounding event loci in the ancestral and descendant sequences, showing event initiation may be modulated by poly(dA:dT) tracts which in turn cause an increased propensity for DNA bending and DNA double-stranded break (DSB) formation. Finally, we explore associations between event loci and human-specific genomic landmarks, including features involved in transcriptional regulation.

## Results and discussion

### PairHMMs for detecting template switch events

To model sequence evolution according to just single base substitutions and indels, and sequence evolution which additionally incorporates template switch events, we implemented two probabilistic models: a canonical three-state pair hidden Markov model (pairHMM) for

linear pairwise sequence alignment, and a seven-state pairHMM-like model that additionally incorporates a single region of reverse complement alignment which corresponds to a candidate template switch event.

PairHMMs are probabilistic models that emit a pair of aligned sequences given two input sequences $x$ and $y$ which, in the context of DNA sequence alignment, consist of nucleotides $x_1, \ldots, x_i, \ldots, x_n$ and $y_1, \ldots, y_j, \ldots, y_m$ [25]. For each alignment column, the probability of emitting a particular pair of symbols is given based on the model state, determined at each column according to the distribution of transition probabilities in the previous state, and the emission probability distributions for that state of either a match/mismatch (emission of the pair $[x_i, y_j]$, called a match when $x_i = y_j$ or a mismatch when $x_i \neq y_j$), or a gap in one sequence ($[x_i, -]$ or $[-, y_j]$). Probabilities are typically transformed into log-space for convenience, and the total probability of the alignment is the sum of the log-probabilities of each alignment column, yielding a global probability value for the most probable path through the pairHMM.

In a typical nucleotide (nt) sequence alignment, sequence homology under a pairHMM alignment is the alternate hypothesis, and the null hypothesis of no sequence homology may be rejected by comparing the global pairHMM alignment probability to that of a null alignment model in which the two sequences are emitted independently of each other [25]. The occurrence of a single template switch event is our alternative hypothesis, and the null hypothesis is that no template switch event was involved in the creation of the descendant sequence. The null hypothesis may be rejected by comparing the probability of an alignment generated under a model that emits linearly aligned sequences solely through substitutions and indels, to that of a model that emits an alignment consisting of substitutions, indels and a single template switch event. We briefly describe our implementation of these models below, but see Methods and S1 Algorithms for full details of both.

The first model, a three-state pairHMM (Fig 2A), defines the probability of an alignment of two sequences that evolved undergoing only substitutions of individual nucleotides and indels. This is a standard approach for the probabilistic alignment of two biological sequences [25], and we refer to this as a unidirectional pairHMM. The second model is formulated similarly to a typical pairHMM (Fig 2B); it consists of seven hidden states, each of which emits a pair of aligned nucleotides, and the probabilities of transitioning out of each state sum to 1 (Methods, Eq 1). Because this model is a compilation of three pairHMMs, with a period of reverse complement alignment in state $M_2$, and requires three combined recursions to fully decode the state path (see Algorithm B in S1 Algorithms), it cannot be considered a true pairHMM as classically defined by [25]. A more general description could perhaps be achieved by formulating our model as an alignment-constrained pair stochastic context-free grammar, such as those used for RNA gene structure and prediction [26–28]. However, given the similar statistical properties and convenient terminology provided, we opted to describe our model using a pairHMM formulation, and refer to this model as a template switch alignment pairHMM (TSA pairHMM).

The TSA pairHMM defines the probability of alignments of sequences that evolved not only undergoing substitutions and indels, but also a possible single template switch event (Fig 2B). This model can be considered bidirectional with respect to the ancestral sequence, capturing a single period of alternate templated replication that proceeds in reverse orientation (see Fig 1A, green arrows), emitting reverse complement sequence for the ② → ③ region. To reconstruct a candidate template switch within each alignment, we wanted the model to decode the single set of switch point coordinates with the highest probability from the TSA pairHMM state path. We therefore use the Viterbi algorithm [29] to infer the optimal path through the unidirectional pairHMM, and a Viterbi-like algorithm consisting of three recursions to infer the optimal TSA pairHMM state path (see S1 Algorithms). Note that choosing a decoding algorithm which only evaluates the most probable state path could cause a loss of

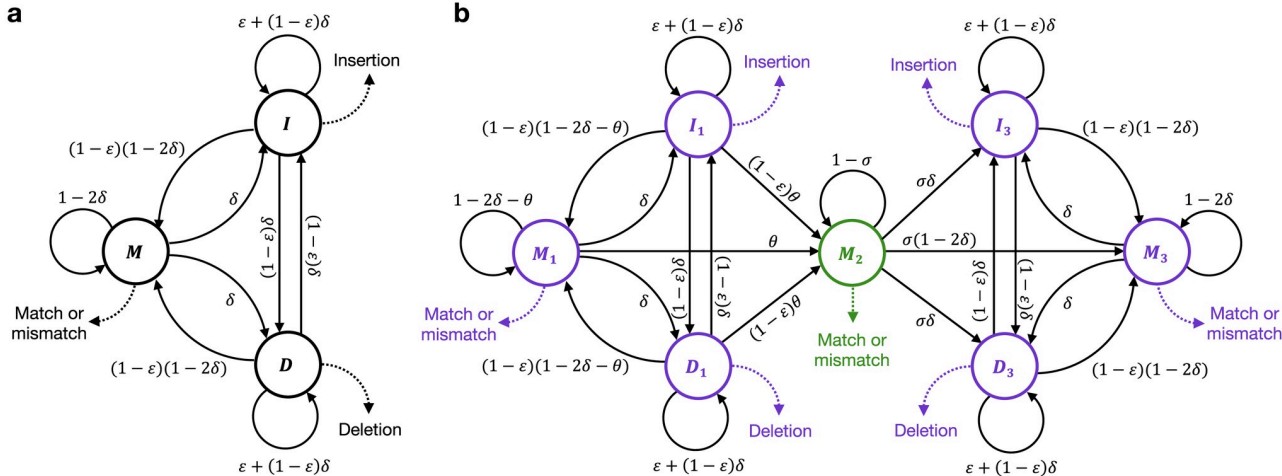

**Fig 2. Models for unidirectional alignment and for (bidirectional) template switch alignment.** (A) The unidirectional pairHMM. The model's three states, $M$, $I$ and $D$, represent respectively match/mismatch, insertion and deletion alignment columns. A match/mismatch ($M$) column is one where both sequences have a non-gap character; an insertion ($I$) column has a gap character ($-$) in the ancestral sequence; and a deletion column ($D$) has a gap character in the descendant sequence. The pairHMM graph illustrates the probabilities that one type of column follows another in a pairwise alignment, with $\delta$ and $\epsilon$ representing gap opening and extension probabilities. For example, the directed edge from state $M$ to state $I$, annotated with $\delta$, denotes that the probability that an $I$ column follows a $M$ column is $\delta$. Dashed arrows represent emissions (the observations of specific alignment columns given the corresponding state); for example, at an $M$ column the two sequences can be either identical ("Match") or contain different nucleotides ("Mismatch"), and one nucleotide from each sequence is emitted in this case. (B) The template switch alignment pairHMM. States $M_1$, $I_1$, $D_1$ emit fragment ⓛ → ①; state $M_2$ emits fragment ② → ③; and states $M_3$, $D_3$, and $I_3$ emit fragment ④ → ⓡ. Parameters $\theta$ and $\sigma$ control the probabilities of template switch initialisation and extension, respectively. Purple states align forwards with respect to both sequences, whereas the green state aligns the two sequences in opposite directions. Emissions in state $M_2$ differ from $M_1$ and $M_3$ in that the emitted sequence respects the complementarity of the alternative template rather than a direct match between the two sequences at that position.

power when inferring candidate template switches with ambiguous switch point coordinates. However, our subsequent focus on establishing and applying stringent probabilistic thresholds to candidate events supports the use of Viterbi/Viterbi-like algorithms here.

The best alignment under each model is therefore represented by the path with the greatest probability; that is, under the assumptions of no template switch and of a single template switch, respectively. We use the logarithm of the ratio of these two probabilities (LPR) as a test statistic for computing a measure of significance for the alternative hypothesis that a single template switch event occurred between input sequences $x$ and $y$, providing a suitable significance threshold is established for this LPR. For each candidate event described below, LPRs are calculated using a sequence region containing one mutation cluster and ±40 nucleotides either side of the mutation cluster. Throughout, "short-range" refers to a template switch detected within this region ±100 nucleotides; for further details, refer to the description in S1 Algorithms and S10 Fig.

## Simulations of template switching to determine a significance threshold for individual events

We sought to establish a threshold on the LPR between the two generated alignments that maximises the recall of true template switch events and minimises the number of false positives caused by erroneously explaining a true cluster of substitutions and indels as an apparent template switch.

To this end, we realistically simulated sequence evolution for human, chimpanzee and gorilla, and applied our alignment models to these simulated data. We simulated two types of evolution: first, without template switches (but with substitutions and indels) so as to

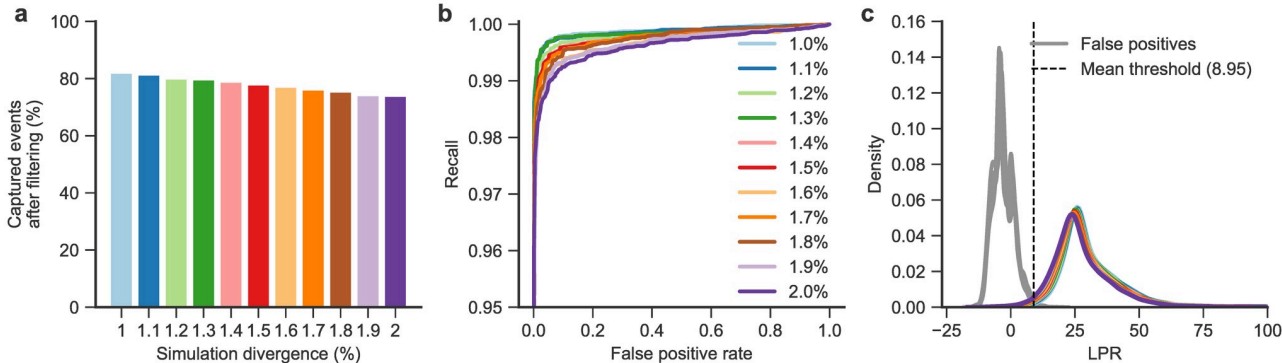

**Fig 3. Distinguishing simulated events from background mutation clusters and setting alignment quality thresholds. (A)** Percentage of events recaptured from simulations of template switch events alongside substitutions and indels using INDELible across a range of divergences. **(B)** Receiver operating characteristic (ROC) curves for discriminating between simulated template switch events and background mutation clusters. Simulations using divergence $t$ from 1–2% in 0.1% steps are shown ($t$ value for each curve indicated by matching colour in part A). Note that the $y$-axis begins at 0.95 for clarity. **(C)** Density curves of LPRs for true positive (i.e. intentionally introduced) template switch events in colours corresponding to (A), and false positive events across all simulation values of $t$ (background/chance mutation clusters) in grey. The mean LPR threshold required to achieve a FPR of 0.005 across simulations is shown as a dashed line.

determine the false positive level of our methods depending on the chosen LPR threshold. Second, we simulated sequence evolution in the presence of template switches, to investigate the detection power we can hope to achieve and assess the false negative rates for given thresholds (see Methods: Sequence simulations).

Even at small evolutionary distances, many template switch events are obfuscated by surrounding neutral mutations, allowing us to capture an average of 78% of introduced events when simulating between 1–2% divergence (Fig 3A). Of the recaptured events, a threshold on the LPR is able to successfully discriminate between true positives (introduced events) and false positives (background mutation clusters) (Fig 3B). Setting a false positive rate of 0.005 still enables a high average recall (0.85±0.04 SD across simulated divergences) of recaptured events, achieved at an average LPR threshold of 8.95 (Fig 3C). For subsequent analysis, we set our LPR threshold to 9, forming our significance cutoff for rejecting the null hypothesis that no template switch event was involved in the creation of an aligned descendant sequence. This threshold is fixed across pairwise comparisons to assign the same level of significance to all detected hominid events. Simulations at smaller evolutionary distances provide a modest improvement in recall (Fig 3B). Divergence in both pairHMMs is specified using the parameter $t$ (see Methods) which, for each simulation, we set equal to the corresponding parameter value used with INDELible to represent the simulated evolutionary distance. We confirmed that our inferences are robust to misspecification of $t$ (see S1 Fig). While our method is able to detect template switch events in a robust manner, it is worth reflecting on the observation that sequence evolution can rapidly obfuscate the signal from past template switch events. Even when simulating at small evolutionary distances of 1–2%, we see that simulated events are often not recaptured due to background substitution and indel processes overlapping the event region (Fig 3A), and additional events are detected but are obscured (falling under the LPR cut-off in Fig 3C). This suggests that short-range template switching is likely more prevalent in the evolutionary history of the hominids than our model is able to detect.

## Template switch events are prevalent in the genomes of great apes

We applied our model to whole genome pairwise alignments between human/chimpanzee, human/gorilla and chimpanzee/gorilla in regions that contain identified mutation clusters,

considering each species ancestral and descendant in turn for each pair (see Methods). For each pairwise comparison, *t* was appropriately set in the model and events were removed from the event set if either the LPR was non-significant or if one of the additional filters was not passed. After this procedure, 4017 significant events were identified across the six comparisons. Unidirectional and TSA pairHMM alignments for all significant events are provided in S1 Data, and the corresponding human genome (GRCh38.p12) coordinates of the mutation clusters associated with each event are provided in S2 Data.

With these significant events identified, accurately placing each event onto the hominid tree and determining their evolutionary direction (see Methods) is desirable for several reasons. It increases confidence in events we identify as significant, as events for which an unambiguous direction cannot be established either reside in regions of poor assembly quality in one or more of the target genomes or of poor multiple sequence alignment, or are obscured by the co-occurrence of background mutational processes. It also enables the assignment of an event type (the ordering of switch point locations with respect to the ancestral sequence; see below) to each unique event, allowing us to infer whether each one could have arisen via intra-strand template switching or inter-strand template switching. Finally, knowing the ancestral and descendant sequences allows us to investigate potential causative ancestral, and consequent descendant features associated with events.

Accounting for poor assembly quality, incomplete lineage sorting (ILS) [30], and "event reversibility" (see Methods and S2 Fig), we successfully placed almost all significant events on the hominid tree (Fig 4). Only 6 events remain unresolved (Fig 4, black bars), representing either regions of poor alignment quality or false positives which marginally pass the LPR threshold. Of the resolved events, 1310 are consistent with the species tree and significant across all expected pairwise comparisons (Fig 4, dark blue bars, dark blue dots); 193 are consistent with a pattern of ILS and are significant across all expected pairwise comparisons (e.g. human appearing ancestral to both chimpanzee and gorilla, Fig 4, dark blue bars, teal dots); 125 are significant across appropriate comparisons but could either be consistent with the species tree or with ILS, and cannot be unambiguously placed on a branch without additional outgroup comparisons (Fig 4, dark blue bars, red and brown dots); 2170 are consistent with either the species tree or with ILS, but are not significant across all expected comparisons (Fig 4, light blue bars); and 213 cannot be placed on the hominid tree due to a missing or entirely gapped alignment block in one comparison (Fig 4, grey bars, grey dots). Among these event classes, it is likely that the most prevalent—those detected in an evolutionarily consistent set of comparisons, but not significant across all comparisons—is due to event obfuscation through background mutation accumulation in event regions, as demonstrated by our analysis of simulated event sets (Fig 3A).

For the purposes of subsequent analysis, we define two event sets of interest. First, the "unique" event set contains all 4017 of the significant events outlined above, allowing us to compare events discovered using our approach to that of [15]. Second, the "gold-standard" subset comprises events that are consistent with the species tree or with ILS and are significant across all relevant pairwise comparisons, allowing unambiguous placement on the hominid phylogeny (*n* = 1503; Fig 4, dark blue bars, dark blue and teal dots). It is worth noting that while we emphasise confident placement of events onto specific branches for the gold-standard set, many significant events inferred with a high LPR are harder to place unambiguously because they are reversibly detected (see Methods) but could be considered gold-standard if a more complete great ape phylogeny was used to facilitate lineage assignment. We use the gold-standard events to investigate genomic features associated with events' ancestral and descendant sequence contexts and physical properties of DNA surrounding event loci.

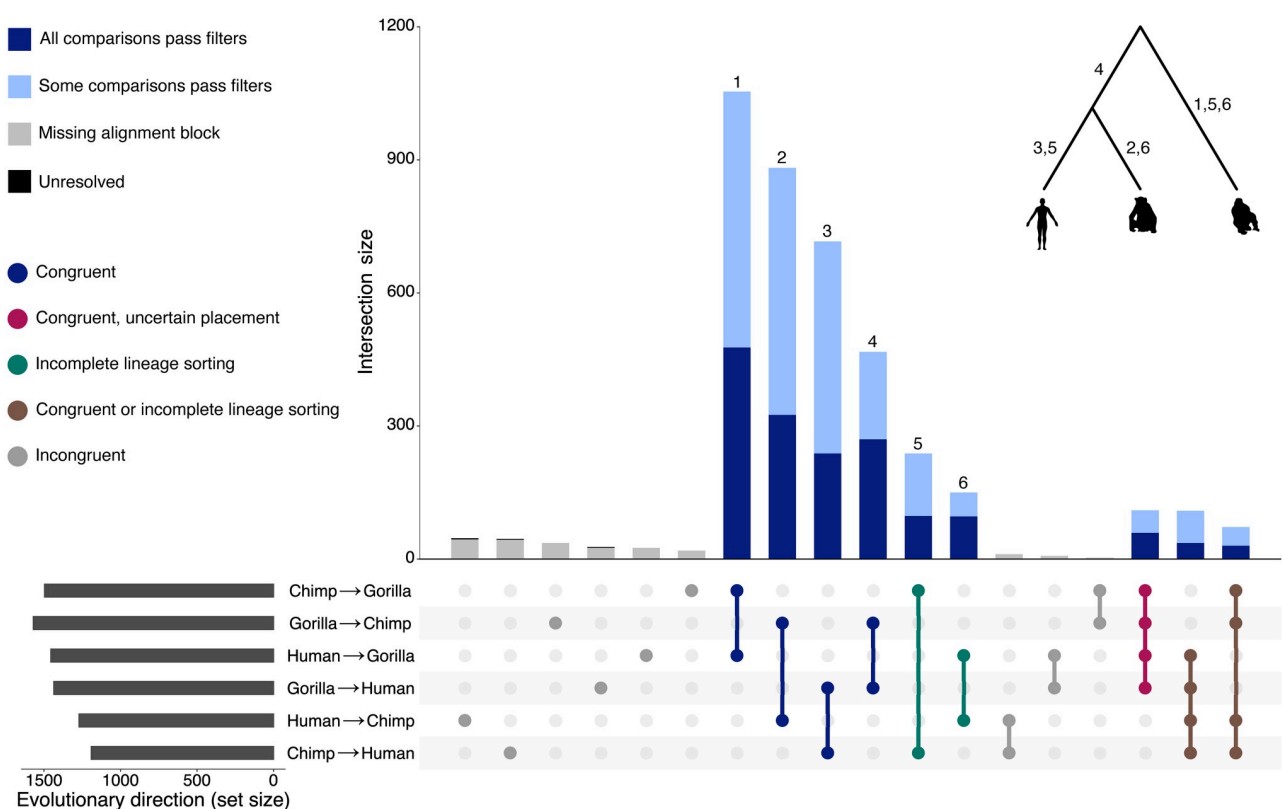

**Fig 4. Evolutionary direction of events.** For each of the 4017 unique events, the intersection of pairwise genome comparisons in which it was found is indicated by the columns of bold/connected circles in the dot matrix, with corresponding intersection sizes shown above as the vertical bar plot. Detected event set sizes for the six pairwise genome comparisons are shown on a horizontal bar plot. Intersections in the dot matrix are coloured according to expected direction: dark blue represents consistency with the hominid species tree, grey intersections should not be observed, teal represents incompatibility between the local tree and species tree consistent with ILS, red represents consistency with the hominid tree but uncertain branch placement, and brown represents events that are consistent either with the hominid tree or with ILS and cannot be resolved without further outgroup comparisons. Counts of evolutionarily consistent events that pass all filters are shown as dark blue bars, events with a consistent set of directions for which one or more of the comparisons has a non-significant LPR or fails an additional filter are shown in light blue, and events for which one of the genomes in this region is either absent from the alignment block or entirely gapped are shown in grey. A total of 6 events with an unresolved direction are shown in black at the top of the grey columns for human→chimp, chimp→human and gorilla→human comparisons; these are near-invisible due to their small numbers. Numbers above the bars of each consistent direction set indicate unambiguous placement of those events on the correspondingly numbered branch of the displayed hominid phylogeny.

We assessed how our method compares to that of Löytynoja and Goldman [15] in terms of the number of events confidently detected, and the impact of our replacement of some non-probabilistic filters with probabilistic thresholds and statistical tests. After performing the same analysis as above but using their model and filtering scheme, we identified 3056 unique events across the three sets of pairwise comparisons (S3 Fig). Despite our larger unique event set, the number of events with an "unresolved" evolutionary direction drops from 8% (246/3056 unique events) using their approach, to 0.15% (6/4017 unique events) using our approach (Fig 4). This demonstrates that our methods are superior in terms of both the total events recovered from pairwise alignments between closely related species and capability to interpret this larger set of events in their phylogenetic context.

Short templated insertions are the most difficult class of rearrangement to capture in an evolutionary context, as many will plausibly present as a mutation cluster or short indel event in a multiple sequence alignment. Focusing on the gold-standard event set, our model largely captures and confidently explains such short templated insertions in the hominids whilst

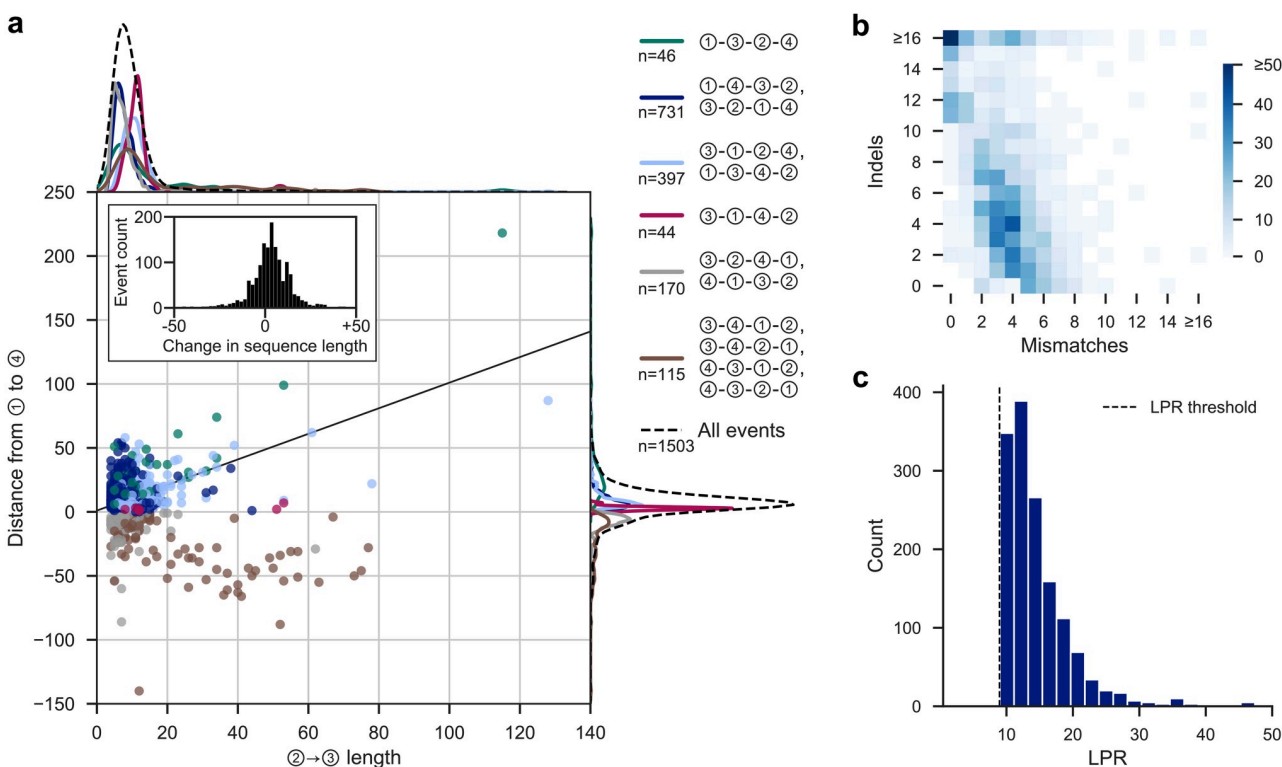

**Fig 5. Summary statistics for template switch events in the gold-standard set. (A)** Comparison of ②→③ lengths and the corresponding ①→④ distances for the gold-standard events. The line $y = x + 1$ corresponds to no net change in sequence length. The inset histogram shows the change in length between the pre- and post-event sequences. Points' colours correspond to event types (legend, right), with the same colours used to show marginal densities at the top and right of the plot (see also S4 Fig). The marginal densities for all gold-standard events (black dashed lines) are drawn on an enlarged scale, for clarity. **(B)** Composition of the template switch-generated mutation clusters in the unidirectional alignments in terms of mismatches and indels. Axes are capped at 16 for clarity. **(C)** LPRs of gold-standard events. The x-axis is capped at 40 for clarity; note that 60 events have a LPR greater than 50. The LPR threshold of 8.95 (Fig 3C) is shown as a dotted line. All summaries are derived from the 1503 events which comprise the gold-standard event set, randomly choosing the output of one pairwise comparison per event.

maintaining the ability to capture longer templated insertions (Fig 5A, median ② → ③ length = 12, median absolute deviation (MAD) = 4.5, max = 128; see also S4 Fig). Few gold-standard template switches leave sequence length unchanged in the descendant species; 65.0% of events increase the length of the post-event sequence, 29.5% decrease the length, and 5.5% cause no net change in length (Fig 5A). Mutation clusters in the input linear alignments which are attributed to these events generally contain more than the minimum of two base differences required to initiate a template switch alignment (Fig 5B, median of 10 differences per cluster, MAD = 4.5). Template switch events therefore plausibly explain thousands of mutation clusters and short indel events across the hominid tree that would previously have had either an incorrect or no attributed generative mechanism. The LPR distribution for these alignments indicates high numbers of events falling at the lower LPR values (Fig 5C), suggesting that if the LPR threshold was relaxed slightly from our conservative choice, the number of unique events discovered could increase considerably. Additionally, many events that are not significant across all comparisons (Fig 4, light blue bars) fall only marginally below the LPR threshold due to our heavily penalisation of substitutions in the model, meaning post-event substitutions may have caused non-significance in one or more pairwise comparisons. We did not attempt to relax thresholds to capture more events as significant, as limiting the false positive rate in our gold-standard events was our primary aim for downstream analyses. However, combined with the

demonstrated inability of our approach to recapture events that are obfuscated by too many additional background mutations (as in our simulations, Fig 3A), we further suspect that the overall rate of template switching in hominid genome evolution is greater than reported here.

For each event, the ordering of the four switch points facilitates the description of post-event rearrangement patterns and the inference of intra-strand and/or inter-strand switching. Following Löytynoja and Goldman [15], we use the four relative positions to ascribe an "event type" to each event: for example, the event type in Fig 1B (bottom) is ①-④-③-②, based on the linear order of the switch points projected onto the ancestral sequence. As we have resolved the evolutionary direction of all events in our gold-standard set, we are able to accurately infer event types and their associated rearrangement patterns. In addition, with a direction-resolved event type defined for each template switch, we are able to infer if an event could have arisen through intra-strand switching, inter-strand switching, or either. This follows the simple logic that for events to arise through intra-strand switching, point ② must precede point ① in the ancestral sequence; if instead ② is located ahead of point ① in linear sequence space, the necessary nascent strand has not yet been synthesised and cannot facilitate an intra-strand template switch. We observed many events that can arise through both intra-strand and inter-strand switching (S1 Table), and the majority rearrangement patterns (①-④-③-② and ③-②-①-④) generate single inverted repeats (as in Fig 1). We also identified many events in which point ④ precedes point ①. Whatever the precise rearrangement mechanism, under the four-point model these events require that the newly synthesised DNA double helix is opened to facilitate the return switch event from point ③ to ④ in a manner conceptually consistent with strand invasion followed by displacement-loop formation in break-induced replication [18]. These rearrangements tend to appear as a single, large insertion in the unidirectional alignment (e.g. S5 Fig), meaning the approach of [15] cannot capture these events as the template switch alignment was required to contain at least two fewer mismatches than the corresponding unidirectional alignment. Our approach of assessing significance through log-probability comparisons allows us to omit this filter and facilitates the capture of significant events that display these viable rearrangements.

As well as being unable to detect these '④ before ①' events, Löytynoja and Goldman [15] assumed chimpanzee represents the ancestral state for every event they detected in the human genome. This assumption is incorrect (Fig 4) and therefore led them to erroneous event type inferences. These methodological artefacts led to other inferences that we now overturn, namely that template switch events appear to occur solely via inter-strand switching and that the generation of a single inverted fragment through ①-③-②-④ events was the most common event type [15].

Using a fully probabilistic approach for template switch event discovery has enabled the identification of ~30% more significant and evolutionarily consistent events than an approach based on a constant scoring scheme coupled with conservative filtering, and has allowed us to assign statistical significance values to events in the final event sets. In addition, defining a gold-standard subset with fully resolved evolutionary directions has allowed us, for each event, to correctly define the ordering of switch points with respect to the ancestral sequence and infer the rearrangement pattern present in the descendant sequence. Using this larger set of significant events with resolved directions, we can better assess associations between event loci and a variety of genomic features in both the ancestral and descendant species.

## Human genomic elements associated with event loci

To investigate associations between functional genomic elements and event loci, we focused on the human coordinates of our gold-standard events, allowing us to use human-specific

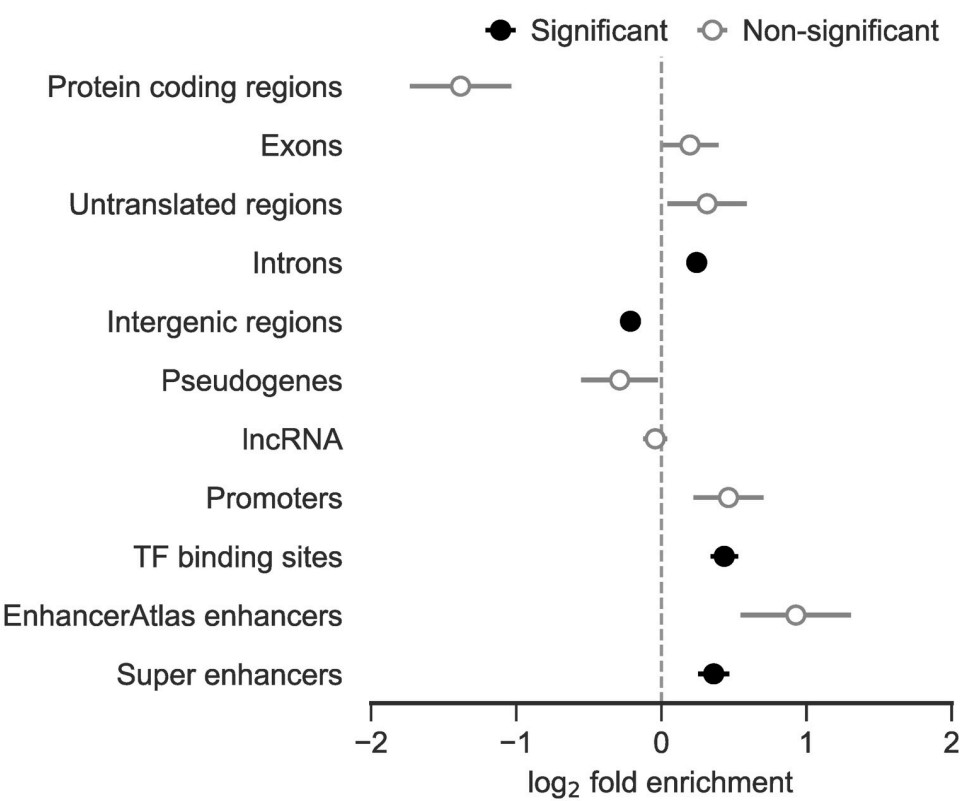

**Fig 6. Enrichment or depletion of gold-standard events within various human genomic elements.** Error bars reflect standard deviations of the $\log_2$-fold changes from each test. A significance threshold was set at 0.01 for Bonferroni-corrected empirical $p$-values.

genomic annotations and experimental data. We found a significant enrichment ($p < 0.01$) of events within introns, transcription factor (TF) binding sites, and super enhancers (Fig 6; see Methods). It is unsurprising that events occur preferentially within introns whilst being depleted in protein coding regions, in line with purifying selection creating mutation intolerant regions. More interestingly, the enrichment of events within features involved in transcriptional regulation suggests that some of the gold-standard template switch events captured here may have contributed to the previously observed high rates of TF binding site and enhancer turnover [31].

As the apparent mutation clusters generated by single template switch events could generate a signal of species-specific accelerated evolution, we additionally checked whether any of the event-associated mutation cluster coordinates intersected with human and primate accelerated regions [32–36] (see Methods). We found 5 events from the unique set within human accelerated regions, and 11 events within primate accelerated regions (1 and 5 events, respectively, from our gold-standard set; S3 Data). While this makes it clear that template switch events are not responsible for the majority of mutation patterns interpreted as accelerated regions, the detected overlap does demonstrate that caution required in their interpretation, as complex mutation patterns generated by either a single short-range template switch or a larger scale mechanism of structural variant formation may generate a signal similar to that of lineage-specific accelerated evolution by multiple substitutions and small indels.

## Physical properties of DNA in the vicinity of gold-standard event loci

Focusing on more local sequence features, the physical properties of the DNA duplex such as thermodynamic stability and localised flexibility have been shown to modulate template switch-mediated structural variant formation in larger scale mutational mechanisms [8, 38]. To investigate any such biases which may underlie short-range template switch events, we use our gold-standard event set to analyse the relationship between event loci, physical properties and local sequence biases.

DNA sequences capable of adopting stable secondary structures such as hairpins are prevalent throughout eukaryotic genomes. These structures are particularly prone to form when DNA is exposed as a single strand during replication, and once formed can cause fork stalling and strand dissociation [39]. We therefore investigated whether the initiation of template switches at ① is biased by local DNA secondary structure stability. A 50nt sliding window was utilised to calculate GC-adjusted minimum free energy (MFE) DNA secondary structures in regions ±500nt around position ① (Fig 7A; see Methods), focusing on ① as we assume any local genomic features will be associated with the site of the initial switch event. We observed two interesting signals of secondary structure stability within these regions. First, secondary structures are significantly less stable in regions flanking ① for both the ancestral and descendant sequences compared to a random genomic background (Fig 7A, $p < 2 \times 10^{-16}$, Wilcoxon rank-sum tests). This may be a residual effect of the greater AT content in these regions compared to the random genomic sample (S6 Fig), as the A:T base pair is less thermodynamically stable than C:G [40]. Second, there is a striking increase in descendant secondary structure stability in the immediate vicinity of ①, and a smaller but noticeable increase in ancestral secondary structure stability across similar positions (Fig 7A). It is unsurprising that we observe such stable structures in the post-event descendant sequences, as the template switching process implicitly generates regions of nearby perfect inverted repeats (e.g. Fig 1B) which are prone to forming the hairpin and/or cruciform structures that constitute highly stable DNA secondary structures [41]. In the ancestral sequences, the smaller decrease in observed free energy around ① is reflective of pre-event potential for structural formation in a subset of events, suggesting that some events may involve hairpin-mediated quasipalindrome-to-palindrome conversion as in the original mechanism proposed for bacteria [22]. Regardless of ancestral stability, the spontaneous creation of sequence regions capable of forming stable secondary structures is of note, as small regions of stable structure play a role in several biological processes [42, 43], and regions of similarly stable structure can cause fork collapse, DSB formation and trigger genome instability [44, 45].

Regions capable of forming stable secondary structures within AT-rich sequences are abundant across chromosomal fragile sites throughout the human genome and typically display increased DNA duplex flexibility [46]. In addition, increased duplex flexibility is observed immediately at the breakpoints of some large-scale mechanisms of structural variant formation in the human genome [38], and we suspected that atypical patterns of flexibility may be observed at event loci. Using our gold-standard events, we investigated measures of flexibility centred on switch point ①, focusing on helical twist, propeller twist, and minor groove width. Helical twist angle, a measure of the inter-bp rotations with respect to the helical axis, is significantly greater in both the ancestral and descendant sequence regions surrounding event loci ($p < 2 \times 10^{-16}$, Wilcoxon rank-sum tests), with a spike immediately around switch point ① (Fig 7B). We also observed a significant decrease in propeller twist, a measure of the inter-bp plane angles, in the vicinity of event regions ($p < 2 \times 10^{-16}$), with an increase at switch point ① that does not reach parity with genome-wide mean values (Fig 7C). Deviations in propeller and helical twist values from those of B-DNA is indicative of DNA bending [47]. Interestingly, DNA bending has been shown to facilitate the error-free template switching DNA damage

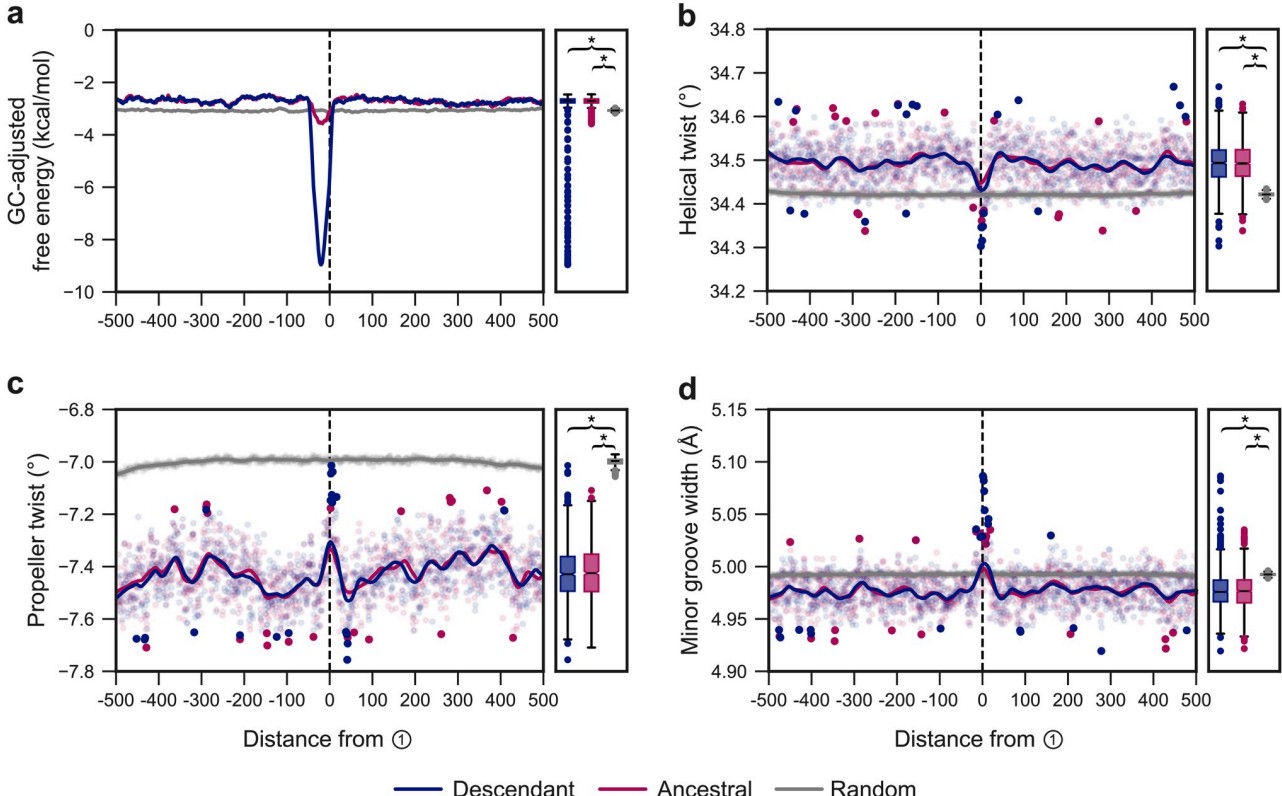

**Fig 7. Single-nucleotide resolution signals of DNA secondary structure stability and bendability for the gold-standard event set. (A)** Mean GC content-adjusted free energies of the MFE secondary structures for the ancestral and descendant sequences, compared to a random genomic background ±500nt around switch point ① using a left-aligned sliding window size of 50 in single nucleotide steps (e.g. at position -500, the MFE structure is calculated using the sequence from position -500 to -451). Marginal box plots summarise the distributions of mean values within the ±500nt region, and brackets indicate significantly different MFEs ($p < 2 \times 10^{-16}$) between groups under a Wilcoxon rank-sum test. **(B, C, D)** Mean predicted helical twist, propeller twist and minor groove width ±500nt around switch point ①. Points represent mean feature values as calculated using DNAShapeR [37], utilising a pentamer sliding window centred on each position, and a Loess fitted curve is overlaid. Additionally, the smallest and greatest 1% of mean values are shown as solid points to highlight extreme values. Box plots as in (A).

tolerance pathway in yeast, facilitated by the high mobility group protein Hmo1 [48]. While distinct from the process we model here, the mechanistic similarity between these local template switch mechanisms coupled with our predictions of non-B DNA values of helical and propeller twist suggests that a propensity for DNA bending may indeed have helped facilitate events in our gold-standard event set.

Lastly, we also observed a more narrow minor groove in the flanking regions around ① compared to the genomic background level (Fig 7D). Decreased minor groove width has been shown to confer resistance to DNA damage by limiting accessibility of the DNA to reactive oxygen species [49, 50]. It is conceivable that a widening of the minor groove, as observed immediately at ①, may likewise cause increased rates of DNA lesion formation that can be bypassed by a template switch process to restart a stalled replication fork, but it is difficult to confidently draw this conclusion without supporting experimental observations.

## Regions surrounding template switch events are enriched for poly(dA:dT) tracts and AT-rich sequences

The structural features identified around event loci consistently show the hallmarks of AT-rich and poly(dA:dT) tract DNA, which are associated with large negative values of propeller twist

and a narrowing of the minor groove [51]. To identify the prevalence of poly(dA:dT) tracts, and any additional sequence motifs which may contribute to event formation, we searched for significantly enriched DNA motifs in a region ±150nt around switch point ① in the ancestral sequences of the gold-standard event set. Using three ranges of motif size (6–10nt, 10–20nt, 20–50nt), we identify the significant enrichment of several A and T dominant motifs across all tested motif sizes (S7 Fig). The most significantly enriched motifs of each size are $T_8$ (183 events, $E = 7.5 \times 10^{-159}$), $T_{10}$ (101 events, $E = 4.8 \times 10^{-46}$), and $YT_2YT_{21}$ (102 events, $E = 6.1 \times 10^{-18}$). In no tests did a motif with greater sequence complexity appear as more significantly enriched than AT rich sequence alone, suggesting that poly(dA:dT) tract DNA plays a more important role in event initiation than any more complex template switch associated motif. It is well established that such poly(dA:dT) tract DNA consisting of ⩾4–6 consecutive A:T base pairs causes intrinsic bending of the DNA molecule [52, 53]. Supported by our predictions of increased flexibility around ① in our gold-standard event set (Fig 7B and 7C), we suggest that sequence-directed bending of the DNA molecule may occur around the initial switch event, similar to that of Hmo1-mediated bending in DNA damage tolerance template switching pathway in yeast [48]. In addition, poly(dA:dT) tracts are known sites of preferential fork stalling and collapse due to elevated rates of DSB formation [54]. The enrichment of these motifs supports the notion that short-range template switching may either be involved in fork restart during DNA lesion bypass, or may occur post-replication in a similar fashion to large-scale structural variant formation in the presence of DSBs caused by persistent lesions unresolved by repair pathways [8].

In combination, the sequence biases and physical properties surrounding event loci indicate that the gold-standard events captured by our model preferentially occur in regions that are prone to replication stress, as previously outlined for well-established mechanisms of larger scale structural variant formation [8]. This validates the events identified as significant using our approach, and confirms that our method provides a previously unachievable resolution in the capture and description of small-scale replication-based rearrangements in their evolutionary context.

## Conclusion

We have identified thousands of significant template switch-mediated mutations across the great ape tree, demonstrating the power of pairHMMs for confidently detecting a class of rearrangements which are traditionally difficult to model. By capturing and assigning an evolutionary direction to many of these events, we are able to explain the presence of thousands of short indels and complex mutation clusters in the evolutionary history of the hominids. Our approach appears robust to selected parameter values, and represents a methodological improvement over a previous non-probabilistic method [15] for modelling short-range template switch mutations in an evolutionary context. By shifting to probabilistic thresholds and assigning statistical significance to individual events, we have achieved superior recall and a consequent improvement in statistical power for identifying associated genomic features.

A limitation of our method is that many events that are characterised by the conversion of a near-perfect inverted repeat into a perfect inverted repeat are classed as non-significant. This quasipalindrome-mediated mutational pattern is the hallmark of a traditional prokaryotic template switch event [22]. However, such events often produce few changes in a unidirectional alignment between the pre- and post-event sequences, in many cases generating solely the minimum of two nucleotide differences that we require to initiate a local realignment under our models. Correcting two-nucleotide differences will not yield a significant LPR, regardless of the length of pre-existing reverse-complement identity (the potential ② → ③

fragment) that the two nucleotides are contained within. While many such mutations may indeed have arisen through legitimate template switch processes, our statistical method cannot report these as robust, statistically supported events in preference to the null hypothesis of simple background mutation. We therefore did not attempt to incorporate these into our final event set, as our priority was to minimise the number of false positive events in our gold-standard event set, rather than maximising the total number of events discovered.

Despite this conservative approach, we have described more events than have been reported previously and can be more confident that the template switches we report represent the true mutational history underlying their associated mutation clusters within linear alignments of the great apes. It is important to emphasise however that care is always required when inferring the mutational history underlying mutation clusters such as those explored here. Other well-characterised mutational mechanisms frequently generate small mutation clusters in eukaryotic genome evolution, such as the multinucleotide substitutions caused by error-prone polymerase activity [55–58]. However, our requirement for a high-homology, reverse-orientation template within 100nt of each focal mutation cluster, coupled with our strict statistical thresholds, demonstrate that a mutation involving a template switch is the most parsimonious explanation for the clusters explored here. We also suspect that the number of events reported here is an underestimate of the true extent to which short-range template switches have shaped the evolution of the hominid genomes.

Our emphasis on reducing false positives has enabled the delineation of physical properties around event loci. It was previously reported that template switch events generate regions with greater energetic potential for DNA secondary structure formation [15], and we have shown this holds in our direction-resolved gold-standard event set. We speculate that an increased potential for fork-stalling secondary structural formation would also be observed in the ancestral species if we did not filter out many of the events involved in quasipalindrome conversion. Nonetheless, it has previously been demonstrated the bypass of stable secondary structures to restart a stalled replisome can be overcome through the recruitment of error-prone polymerases and template switch-mediated DNA synthesis [59]. We therefore suggest that this signal should still be investigated when considering mechanisms which may underlie short-range template switch initiation in future work. More importantly for events identified using our approach, event formation appears to be associated with an excess of poly(dA:dT) tracts which are known replication barriers that can cause fork collapse [54], as well as non-B duplex geometry around event switch points and signals of helical bending which could lead to an increased potential for DSB formation at the initial disassociation site.

A consideration regarding the events we have described here is the signals such rearrangements could create in evolutionary analyses. We identified events both in curated regions of human accelerated evolution [32] and in elements involved in transcriptional regulation which are thought to be subject to high rates of evolutionary turnover [31]. In both cases, observed signals of evolutionary importance, typically interpreted as consequences of a high rate of change, could feasibly be generated by a single complex mutational event such as a template switch. We do not claim that the template switch mutations outlined here underlie regions of accelerated evolution, as we observed few intersections with such regions. However, our observation of some intersections between template switch loci and these regions still demonstrates that care is required when interpreting signatures of high turnover or accelerated evolution.

The short-range template switch events and associated features described in the present work were identified by focusing on local template switching, as it has allowed us to assign enough statistical significance to individual events to distinguish candidate events from accumulated substitutions and/or short indels. While this represents a significant methodological

improvement and the most comprehensive delineation of these events in the hominids to date, it does leave the characterisation of small-scale, non-local template switching unresolved. This will remain the case unless methods for the direct observation of these events are developed.

## Methods

### PairHMMs for modelling the short-range template switch process

We implement two models, a standard (unidirectional) pairHMM (Fig 2A) and the pairHMM-like TSA pairHMM (Fig 2B). Each model is specified by a set of hidden states $H = \{h_1, \ldots, h_N\}$, a transition probability matrix $Q$ with elements $q_{ij}$ representing the probability of moving from state $h_i$ to $h_j$, two input sequences $x$ and $y$ consisting of nucleotides $x_0, x_1, \ldots, x_n$ and $y_0, y_1, \ldots, y_m$, and emission probabilities $s_{h_i}(a, b)$ representing the probability of the pair $[a, b]$ being emitted from state $h_i$ (where $a$ and $b$ indicate nucleotides from $x$ and $y$, or gaps −). We use the symbol $s$ to reflect that the logarithms of such values are often considered as emissions' (additive) s̲cores. For consistency, we represent all our parameters and calculations in terms of probabilities, with logarithms only introduced later (e.g. to create LPRs). In Fig 2, states $H$ are shown as nodes, non-zero elements of $Q$ are shown as directed edges (annotated with the values assigned to them in terms of probabilities $\delta$ and $\epsilon$ as defined below), and non-zero probabilities $s$ are shown as annotated dashed arrows.

The unidirectional pairHMM (Fig 2A) is of canonical form for pairwise alignment [25], composed of three hidden states: match ($M$), insertion ($I$) and deletion ($D$) and, giving $H = \{M, I, D\}$. $M$ corresponds to the emission of a pair of nucleotides $[x_i, y_j]$; no gaps can be emitted. $I$ emits a gap and a nucleotide $[−, y_j]$, and $D$ emits a nucleotide and a gap $[x_i, −]$. State transition probabilities $Q$ are specified using two parameters, $\delta$ and $\epsilon$, where $\delta$ is the frequency of indel events expected along a pairwise alignment and $\epsilon$ controls their lengths. We use $\delta = 1 − e^{-t(\rho/2)}$ and $\epsilon = 1 − 1/\lambda$, where $t$ is pairwise divergence measured in expected substitutions per site, $\lambda$ is mean indel length and $\rho$ is mean number of indel events per substitution. For the purposes of our hominid analyses, we set $t$ based on estimates from [60] and assume $\lambda = 20$ and $\rho = 0.14$ across species comparisons based on estimates from [61]. We have assumed indel lengths are geometrically distributed to allow the use of efficient dynamic programming alignment algorithms, but it is worth noting that a zeta power-law model provides a better description of observed hominid indel lengths [61]. The transition probabilities $q_{ij}$ (Fig 2A) satisfy

$$\sum_{j=1}^{N} q_{ij} = 1 \quad \forall i. \tag{1}$$

Emission probabilities $s$ are defined according to the JC69 substitution model [62], so that equal substitution and indel rates are assumed. It is likely that sequences undergoing template switching violate standard assumptions about sequence evolution regarding base frequencies, GC content and transition/transversion ratio. Using the simple JC69 model of sequence evolution for both of our pairHMMs allows us to account for pairwise divergence when emitting sequences, correctly interpreting the probability of substitutions in each pairwise comparison, whilst foregoing more complex *a priori* assumptions about the evolutionary processes shaping each sequence. Under JC69, the emission probability for states $M$ is given by

$$s_M(x_i, y_j) = \begin{cases} \dfrac{1}{4} + \dfrac{3}{4}e^{-t} & \text{if } x_i = y_j \\[2ex] \dfrac{1}{4} - \dfrac{1}{4}e^{-t} & \text{otherwise} \end{cases} \tag{2}$$

where $t$ is the divergence between the two species in the pairwise comparison. For state $I$, $s_I(-, y_j) = 1/4$ as all inserted nucleotides $y_j$ are assumed to occur with equal frequency, and in state $D$, $s_D(x_i, -) = 1$ to make our alignment conditional on the ancestral sequence (see below). We use the canonical Viterbi algorithm [29] to find the most probable state path through this pairHMM, assuming that the model starts in state $M$ for convenience.

The TSA pairHMM (Fig 2B) has seven hidden states: $M_1$, $D_1$, and $I_1$ which emit alignment fragment Ⓛ → ①, $M_2$ which emits fragment ② → ③, and $M_3$, $D_3$, and $I_3$ which emit fragment ④ → Ⓡ (i.e. $H = \{M_1, I_1, D_1, M_2, M_3, I_3, D_3\}$). The model is structured to capture a single template switch event per alignment by requiring a single transition into $M_2$ from $\{M_1, I_1, D_1\}$ (at ①, ②), and a single transition from $M_2$ into $\{M_3, I_3, D_3\}$ (at ③, ④). As described in Algorithm B in S1 Algorithms, state $M_2$ differs from typical pairwise aligners in that the descendant sequence $y$ is aligned in complement and reverse orientation with respect to the ancestral sequence $x$, capturing the period of alternate strand-templated replication inherent to the template switch process. State transition probabilities $Q$ satisfy Eq 1, and are defined using the parameters $\delta$ and $\epsilon$ from the unidirectional pairHMM and two additional parameters: $\theta$, the probability of initiating a template switch event, and $\sigma$, which controls the expected length of the ② → ③ fragment. We set $\theta = N/(C \times A)$, where $N$ is the expected number of events in a pairwise ape comparison, $C$ is the total count of mutation clusters identified in each pairwise comparison, and $A$ is the event-specific alignment length (see S1 Algorithms for details), and set $\sigma = 1/L$, where $L$ is the expected ② → ③ length. We estimate $N$ as 2750 and $L$ as 10, based on the average number of significant events found in earlier pairwise great ape comparisons and the ② → ③ length distribution of those events.

The precise value of $N$ used likely has little impact: because the product $C \times A$ is large, $\theta$ will always correspond to a small initiation penalty for any reasonable value of $N$. In contrast, $\sigma$ can have a more substantial effect, as this parameter controls the expected length of the ② → ③ fragment. Lower values of $\sigma$ lead to longer ② → ③ fragments being preferred, possibly causing some events to pass (e.g.) the 'all four nucleotides present' filter (see below) and generating some more plausible detected events (e.g. S9 Fig). However, we prefer to use our more natural formulation for this parameter, $1/L$, in pursuit of quality over quantity.

Emission probabilities are set as in the unidirectional pairHMM, so all $M_\bullet$ states follow Eq 2; $s_{I_\bullet}(-, y_j) = 1/4$; and $s_{D_\bullet}(x_i, -) = 1$. While fairly large deletions might truly be explained by a template switch event in which only a short ② → ③ fragment was incorporated between relatively distantly separated points ① and ④, we do not find these deletions convincing under the TSA pairHMM, and we lack a suitable probabilistic model to facilitate their statistical assessment. We therefore opt to effectively disallow such events by setting all $D_\bullet$ emission probabilities to 1 across both models. We further set a threshold of 50 on the maximum number of deletions per template switch alignment, as a candidate event characterised by a single deletion of around this size in the unidirectional alignment is large enough to falsely yield a significant LPR in our analysis.

We use a Viterbi-like algorithm (see Algorithm B in S1 Algorithms) for identifying the most probable template switch alignment, similar in form to that of [63]. In addition to emitting an alignment, our implementation of the TSA pairHMM also outputs annotations to indicate the positions of ①, ②, ③, and ④ (see Fig 1B), given by the indices of transitions into and out of $M_2$ in the state path identified during trace-back. These annotations describe an individual template switch event, and the linear ordering of the four switch points is subsequently used to define event types.

Applying each of these pairHMMs to any two input sequences $x$ and $y$ will produce a score for each model, corresponding to the most probable alignment of $x$ and $y$ under that model.

We generate a test statistic to infer the statistical significance of a template switch alignment (representing an event) over a linear alignment (representing an alternatively explained mutation cluster) by taking the logarithm of the ratio between the most probable alignment scores generated by each pairHMM. This is referred to as the LPR.

## Sequence simulations

We performed simulations to determine a suitable LPR significance threshold to confidently distinguish between short-range template switches and multiple independent substitutions and indels within a small region.

We simulated evolution in continuous time using INDELible [64] under the HKY85 substitution model [65] using nucleotide frequencies calculated genome-wide in human. We used power law-distributed indel lengths, as a zeta power-law model of indel length provides the best fit to indel processes in the hominids [61]. Robust estimates of the evolutionary distance between human-chimpanzee and human-gorilla are in the range of 1.2% to 1.6% [60], and we therefore simulated sequence evolution in 0.1% steps of $t$ between 1% and 2% to cover this range. For each $t$, two types of simulation were performed. The first generated a descendant sequence $y$ given input sequence $x$ by incorporating only substitutions and indels. The second additionally incorporates a single template switch event into $y$. We can then use $x$ and $y$ pairs from each set of simulations as input to our pairHMMs, which we assume will detect our introduced events and produce a distribution of LPRs that can be clearly separated from the distribution of LPRs produced by clustered mutations in the first set of simulations.

In the first set, 24,000,000 bases were simulated by taking as ancestor $x$ 1000 random 1kb fragments from each autosome, as well as chromosomes X and Y, from the human reference genome (GRCh38.p12). We simulated substitutions and indels from $t_0 = 0$ to $t_1$, where $t_1$ is the total divergence, and then globally aligned the original sequence against the simulated descendant sequence using the Needleman-Wunsch algorithm and a simple scoring scheme (match: 2, mismatch: -2, gap: -1) [66].

For the second set of simulations, we want to simulate not only sequence evolution under substitutions and indels, but also under template switch events. To do so, first we select a uniform random time $t_{TS} \in [t_0, t_1]$. We then define a template switch event using the positioning of points ②, ③, and ④ relative to ① from a single high-confidence event randomly drawn from an event set generated between human (GRCh38) and chimpanzee (Pan_tro_3.0), using the model and filtering criteria of [15]. Sequence evolution under substitutions and indels is simulated as before until $t_{TS}$, at which time a uniform random sequence position in the nascent sequence is selected as ① (excluding the first and last 200 bases to guarantee adequate sequence space for the template switch process). The predefined relative coordinates of points ② and ③ are used to source a sequence in reverse complement from the alternative template strand. This sequence is inserted into the sequence in a manner consistent with the template switch process, replacing the nascent sequence between points ① and ④. After this introduced templated insertion, sequence evolution continues as before under substitution and insertion/deletion, from $t_{TS}$ until $t_1$. The coordinates of the introduced event are recorded, and global alignment to the ancestral sequence is then performed.

## LPR threshold determination and filtering for confident template switch discovery

All simulated alignments were scanned with our cluster-identification approach; true positives were defined as detected template switch events introduced intentionally as described above, and false positives are background mutation clusters detected using our model that were

introduced under the first simulation regime. We set thresholds for the LPR that only allow 0.5% of false positive events through.

In addition to LPR thresholds, we require events to pass three further filters. First, the ② → ③ sequence must contain all four nucleotides to prevent low complexity runs missed by masking annotation. Second, events must contain fewer than 50 deletion positions. Finally, we set a baseline alignment quality requirement (see below) for event regions to ensure that we are only modelling regions with reasonable assembly quality, limiting the possibility that our model exchanges one cluster of alignment noise for other, slightly more plausible, alignment noise.

## Template switch events in the hominids

We downloaded the Ensembl (v.98) EPO alignments [24] of thirteen primates, extracting pairwise alignment blocks between human (GRCh38.p12) and chimpanzee (Pan_tro_3.0), human and gorilla (gorGor4), and gorilla and chimpanzee. Gap-only columns were removed for each pairwise comparison, with their positions recorded to allow us to relate the coordinates of events across comparisons to the original multiple sequence alignment coordinates later. To discover events, we considered both species from each pairwise alignment as ancestral and descendant in turn, which facilitates the subsequent placement of events in their evolutionary context (see below). As in [15], we defined mutation clusters within each pairwise comparison as any 10nt window in which two or more nonidentical bases are identified. For each such mutation cluster, the cluster itself and a small sequence region upstream and downstream of the cluster boundaries were considered for alignment (see S1 Algorithms and S10 Fig for further details). This region was aligned using both the unidirectional and TSA pairHMMs, the LPR between the two alignments was calculated, and the statistical significance of the event assessed using the predetermined LPR threshold. In addition to the 50 nucleotide deletion threshold outlined above, we used three additional filters to remove spurious events caused by either low-complexity sequence or alignment regions of poor quality. Low complexity sequences are filtered by requiring that the alternate template sequence which donates the ② → ③ fragment is not masked by RepeatMasker [67]. We additionally require the ② → ③ fragment to contain all four nucleotides; while this may be overly conservative, it removes any concerns about the inclusion of simple di- or trinucleotide repeat expansions in our final event set. Finally, to remove events that marginally improve regions of extreme poor alignment quality, we applied a length-normalised alignment probability threshold (see next section).

## Sampling hominid alignments to determine genome-wide alignment probabilities

To ensure the LPR threshold method is not simply invoking artefactual template switch events in an attempt to correct regions of poor alignment quality or incomplete genome assembly, we used an average alignment quality filter. We sampled 100,000 random 300nt blocks from each of the human/chimpanzee, human/gorilla and chimpanzee/gorilla pairwise alignments. Each block was globally aligned under our unidirectional pairHMM (Fig 2A), with pairwise parameters kept identical to those used for all other analysis. We calculated a length-normalised log-probability for every sampled alignment block by dividing each unidirectional pairHMM alignment log-probability by its corresponding alignment length and set the 20th percentile of the distribution of these values (S8 Fig) as a species pair-specific threshold on the minimum length-normalised log-probability of any template switch alignment. This is assessed for each template switch alignment after subtracting the log-probability contributions of the transitions into and out of $M_2$ from the global event log-probability (low probability events that are not

otherwise allowed for by the 20th percentile threshold). This ensures that template switch alignments in our final event sets are as probable as the majority of linear alignments in the considered pairwise comparisons, rather than just exchanging regions of very poor alignment quality or genome assembly for a comparatively more plausible template switch alignment.

## Phylogenetic interpretation of template switch events

We first identified events which correspond to one another across pairwise comparisons. We converted the pairwise alignment coordinates of each mutation cluster associated with a significant template switch event into their corresponding multiple sequence alignment coordinates and checked for any overlap in the alignment coordinates of each event-associated mutation cluster identified in each pairwise comparison, recording the set of comparisons in which each significant event was found.

Using these sets of overlapping alignment coordinates, we aimed to place each significant event onto its correct branch of the hominid phylogeny. For each pairwise comparison, if the true ancestral and descendant sequences are correctly designated in our model as $x$ and $y$, respectively, and post-event substitutions and indels have not excessively altered the ancestral sequence, the TSA pairHMM is able to reconstruct $y$ from $x$. Assuming these loci are biallelic (presence/absence of a template switch mutation) and assembly quality is high, there should always be two of the six possible comparisons (Fig 4) where the model reconstructs $y$ from $x$. We can use these two comparisons to place an individual event onto the hominid phylogeny. For example a significant event detected in the comparisons with each of the gorilla and chimpanzee sequences, respectively, designated as representing the ancestor ($x$) of human (descendant $y$) is denoted as being found in the gorilla→human and chimp→human comparisons and must have occurred in the human lineage.

However, when considering each species pair as ancestral/descendant ($x/y$) in turn, a subset of events are significant regardless of which species is designated $x$ or $y$, allowing $y$ to be reconstructed from $x$ across four comparisons instead of two as above. We refer to these events as "reversible", and their identification as "reversible detection", as the true ancestral sequence can be reconstructed from the true descendant sequence as well as *vice versa*. An example reversible event is shown in S2 Fig. Event reversibility is determined by the number and length of deletions introduced into the true descendant sequence. For example, if an event causes many deletions in the true descendant sequence $y$, such as a ①-③-②-④ event which replaces a larger region (between ① and ④ of $x$) with a shorter region (reverse complement of ③-② of $x$), too much sequence information will be lost to reversibly reconstruct $x$ from $y$. Adapting our previous example, consider an event that can additionally be detected in both comparisons with the human sequence designated as ancestral. This event is now denoted as gorilla↔human and chimp↔human. From this set of comparisons and directions, we cannot infer whether the chimpanzee and gorilla sequences correspond to the ancestral state (consistent with an event in the human lineage of the species tree), or the human sequence does (consistent with the ILS tree). In such cases, although we observe the event across a consistent set of pairwise comparisons (i.e. we have only observed two possible ancestral or descendant species), we cannot unambiguously place the event onto a single lineage.

Using these methods, we defined an annotation for each set of evolutionary directions across which individual events are discovered (Fig 4, dot matrix and row labels). These annotations are then used to either place events onto individual evolutionary lineages, or to demarcate ambiguous placement when assigning an event to a particular lineage is not possible without further outgroup comparisons. For each unique, significant template switch event that cannot be clearly assigned to either a set of directions which are consistent with the species

tree or with ILS, we investigated the non-significant pairwise comparisons for evidence of template switches that fall marginally below the significance threshold or otherwise fail one or more of the other filters. Unique events that are significant in one comparison, but are either non-significant or fail one of our additional filters are assigned to the appropriate species tree- or ILS-consistent set, but are not used in downstream analyses (Fig 4, light blue bars). Remaining events retain the annotation of incomplete detection (Fig 4, grey bars).

## DNA secondary structure and flexibility

Using our gold-standard events, we calculated physical properties of the DNA duplex to investigate local biases in event formation, focusing on measures of stability and flexibility. For each gold-standard event, the sequence region ±500nt around switch point ① was extracted for the ancestral and descendant sequences. DNA secondary structure prediction was performed using RNAfold v2.4.1 from the ViennaRNA Package [68], using a sliding window of size 50 along these sequence regions and a step size of 1nt. Energy parameters for single-stranded DNA were used, allowing G-quadruplex formation prediction and disallowing lonely (helix length 1) and GU wobble base pairing ("RNAfold - -noLP - -noGU - -gquad - -noconv - -paramfile = dna_mathews2004.par"). For comparison with a genomic background, we randomly drew 10,000 equally sized regions from GRCh38 and performed the same analysis.

GC content heavily impacts the stability of potential DNA secondary structures, as the A:T base pair is less thermodynamically stable than C:G [39]. We therefore regress GC content out of calculated free energies for all MFE structures to identify regions of stable structure independent of underlying GC content. Our sliding window approach assesses sequences of length 50, so an additional G or C nucleotide increases GC content in any window by 2%. Therefore we randomly generated 10,000 nucleotide sequences of length 50 for each possible GC content, 0%, 2%, 4%, . . ., 100%, and calculate the average MFE for each of these set of sequences. The free energies of all MFE structures in the above sliding windows are then adjusted by calculating the GC content of each window and subtracting the the corresponding average GC content free energy as determined using the randomly generated sequences.

We calculated minor groove width, helical twist and propeller twist in these regions, as well as for 100,000 uniform random sampled 1001nt sequences from across all GRCh38 chromosomes, using the DNAShapeR package [45], which is based on the method of [69] for predicting DNA structural information. This approach utilises a pentamer sliding window to calculate each feature as determined through Monte Carlo simulations, accounting for sequence context of the focal nucleotide within the window. As above, this analysis was repeated for 10,000 randomly selected regions from GRCh38 for comparison.

## Motif identification

We generated position weight matrices for significantly enriched sequence motifs using the differential enrichment objective function in MEME [70]. For every event in our gold-standard event set, sequence ±150nt around switch point ① were searched for motifs, in both the ancestral and descendant sequences. If more than one ancestral or descendant sequence was available, chimpanzee and human sequences were used, respectively. Event regions were compared against a global genomic background set of 30,000 301nt sequences, using 10,000 randomly sampled sequences from each of the human, chimpanzee and gorilla genomes, excluding regions containing masked bases or gaps. As we sought to identify individual putative causative motifs per sequence, we allowed one or zero occurrence of each motif per sequence. We repeated this analysis for three ranges of window sizes: 6–10nt, 10–20nt, and 20–50nt, where window size defines the minimum and maximum allowed length of the motif.

The analysis was performed using the command "meme event_sequences.fa -dna -nostatus -mod zoops -minw {6,10,20} -maxw {10,20,50} -objfun de -neg background_sequences.fa -revcomp -markov_order 0 -seed 42". The *E*-value cut-off for significant enrichment was set at $E \leqslant 10^{-6}$.

## Association with human-specific genomic features

We created a set of 13 functional annotations to investigate enrichment/depletion at event loci, as well as processing regions of accelerated evolution in humans from the literature to check for overlaps with events (S2 Table). As indicated in S2 Table, several of the functional genomic annotations were processed using the procedures of [71]. We performed permutation tests to identify enrichment of these features intersecting gold-standard events, using the coordinate of switch point ① from each event to check for overlaps. Background distributions of genomic locations for each feature were generated using randomly selected coordinates from the genomic background of GRCh38, selected using "bedtools random" [72]. We generated 10,000 random sets of coordinates of length equal to the size of the gold-standard event set, disallowing coordinates that fall in GRCh38 gap locations. The log$_2$-fold enrichment is measured with respect to the mean of the genomic background distributions. We determined significant enrichment or depletion by calculating empirical *p*-values as $(r+1)/(n+1)$, using the procedure of [73], where *n* is the number of coordinates in each randomly generated set and *r* is the number of these random sets that intersected with each genomic feature more than the gold-standard event coordinates. We also checked for intersections with human and primate accelerated regions (see S2 Table) in a subset of events for which human was determined to match the descendant state using "bedtools intersect", but did not include this in the enrichment analysis.

## Code availability

C++ implementations of the pairHMMs described here are available from github.com/conorwalker/tsa_pairhmm. Scripts underlying our analyses are available from github.com/conorwalker/template_switching. Figures were created using matplotlib [74], seaborn [75], and UpSetR [76].

## Supporting information

**S1 Fig. ROC curves for discriminating between simulated template switch events and background mutation clusters.** ROC curves for simulations at evolutionary distances of **(A)** 0.005, **(B)** 0.010, **(C)** 0.015, and **(D)** 0.020. At each evolutionary distance, the TSA pairHMM parameter *t* was set independently of the evolutionary distance used for sequence simulation, ranging from 0.001 to 0.02 in 0.001 increments. The ROC curve for the *t* parameter corresponding to the true evolutionary distance is shown as a dashed magenta line, the minimum and maximum fixed *t* values are in dark blue and light blue, respectively, and all other values of *t* are shown in grey. Across all fixed evolutionary distances, almost identical performance is achieved using the true *t* and using the highest fixed value of *t*, while marginally worse performance is observed when fixing *t* to smaller values. The performance differences are so small (as measured by the area under the ROC curve (AUC)) that any misspecification of *t* will have a negligible impact on model performance, indicating that our inferences are robust to our assumed values of *t*. Note that all *y*-axes start at 0.95, as the ROC curves between specified values of *t* would otherwise be indistinguishable.
(TIF)

**S2 Fig. Example of a 'reversible' event.** A cluster of mutations is observed between human/chimpanzee and human/gorilla, appearing as either a large cluster of substitutions (input multiple alignment, top), or as a large insertion and deletion event (unidirectional pairHMM alignments). Regardless of which species is specified as the ancestral sequence *x* or the descendant sequence *y*, the event is detected as significant (reversible detection; S1 Data, event 3803). As we cannot tell whether this event is congruent with the species tree or represents a region of incomplete lineage sorting, we are unable to place it onto an evolutionary lineage. Coordinates are retrieved from the input Ensembl EPO alignment, and in this case refer to positions from sequences aligned to the negative strand of GRCh38. Note that "Anc" refers to the assumed ancestral sequence and "AncC" refers to the complement of this sequence.
(TIF)

**S3 Fig. Overlap between events identified using our approach and the non-probabilistic model of [15], and the achievable resolution of direction for events identified using this previous approach. (A)** Intersection between the set of template switch events found using the approach of [15], denoted "LG17", and the significant set of events identified using the TSA pairHMM. Box plots show log-probability ratios for each event set, as well as for candidate events found with both methods. The *y*-axes are limited to 50 for clarity. **(B)** Evolutionary direction for the LG17 event set; annotation as in Fig 4, but with an additional category in the dot matrix (shown in black, far right), corresponding to events that are not compatible with a three species tree, likely falling in regions of poor quality sequence assembly or erroneous multiple sequence alignment.
(TIF)

**S4 Fig. Summary statistics for template switch events in the gold-standard set: Comparison of 2→3 lengths and the corresponding 1→4 distances.** Plots are exactly as in Fig 5A, with the points and marginal densities for the six distinguishable event types shown on separate panels.
(TIF)

**S5 Fig. Example event in which switch point 4 precedes 1.** Figure shows, top to bottom, annotation, linear alignment, template-switch alignment and underlying switch process. The bold, underlined region between ④ and ① represents the nascent DNA strand prior to the initial switch event at ①, which typically forms hydrogen-bonded base pairs behind the proceeding replisome, preventing its further involvement in ongoing replication. For events in which ④ precedes ①, a direct repeat generated in the descendant sequence (dark blue arrows above the template-switch alignment) indicates that this region was not sequestered from the replisome through base pairing, and facilitated the final ③ to ④ switch event through an open conformation. The mutational consequence of this event is a complicated rearrangement pattern, manifesting as a series of direct and inverted repeats at the sequence level, shown by coloured arrows above the template-switch alignment (direct repeats shown as arrows in the same orientation; reverse complement regions shown with arrows in opposite orientation). This event is number 145, S1 Data.
(TIF)

**S6 Fig. Sequence biases at event loci in the gold-standard event set. (A)** Percentage of each nucleotide in the ancestral and descendant sequence region, compared to a random genomic background. Percentages are calculated in a region ±150nt around ① loci; to form our random background distribution, 10,000 regions of 301nt were randomly drawn from each of the human, chimpanzee, and gorilla genomes. **(B)** Counts of each nucleotide in a left-aligned single nucleotide sliding window of 10 bases, averaged across descendant, ancestral and randomly

sampled sequences at each position.
(TIF)

**S7 Fig. Enriched sequence motifs within ±150nt of switch point 1 for the gold standard events, compared to a random genomic background sampled from GRCh38.** The most significantly enriched motifs (lowest $E$-value; top row) and most frequent significant motifs (bottom row) within ±150nt of ① for gold-standard events. Motifs were tested for enrichment at three motif size ranges: **(A)** 6–10nt **(B)** 10–20nt **(C)** 20–50nt. In (B), note that for the 10–20nt motif search the same motif ($T_{10}$) is both most significant and most numerous.
(TIF)

**S8 Fig. Genome-wide samples of alignment log-probabilities under the unidirectional pairHMM.** The derived log-probabilities of sampled alignment regions are normalised by final alignment length to produce per-base log-probabilities. Dashed lines represent the 20th percentile thresholds used as baseline alignment quality thresholds for event regions for each pairwise comparison. If both the null model and the template switch model alignments in a region fail this threshold, the region is removed from our analyses.
(TIF)

**S9 Fig. Example of an event which is significant and passes all filters when using a smaller value of $\sigma$ than the selected value of $\sigma = 0.1$ used for our inferences.** For the chosen value of $\sigma$ used in the main text (0.1, top), and a nominal small value of sigma ($\sigma = \delta = 0.001$, bottom), an event detected in the human→chimpanzee and gorilla→chimpanzee directions is shown. When using $\sigma = 0.1$, this event does not contain all four nucleotides in the ②→3 fragment, and fails the corresponding filter. If $M_2$ extension is penalized less heavily, by setting $\sigma = \delta$, a longer period of ②→3 alignment is included in the state path during Viterbi decoding, including all four nucleotides and allowing the event to be called as significant. Note that "Anc" refers to the assumed ancestral sequence and "AncC" refers to the complement of this sequence.
(TIF)

**S10 Fig. Diagrammatic overview of how alignment regions are defined in each pairHMM for an example mutation cluster, and how these regions are aligned under each model. (A)** Given an input linear alignment (top), a focal mutation cluster is identified when there are ≥2 substitutions or indel positions within a 10nt window (yellow and red sequence blocks). Mutation clusters vary in their sizes; the 10bp window used for cluster identification is expanded once two differences are found, continuing to expand the rightmost cluster boundary as long as additional differences are found with each iteration of boundary increase. Once a focal mutation cluster is defined (red, yellow), the sequences used for re-alignment are defined separately for each model. **(B)** For the unidirectional pairHMM, the sequence regions defined by the red/yellow mutation cluster in addition to ±40nt flanking sequence (black, from (A) above) are used for alignment. Unidirectional alignment then follows Algorithm A in S1 Algorithms: the figure illustrates initialisation and subsequent calculation of the $M$ matrix of Algorithm A in S1 Algorithms, omitting the $I$ and $D$ matrices for clarity. **(C)** For the TSA pairHMM, in addition to the yellow, red and black regions aligned under the unidirectional pairHMM, a further ±100nt region is included for (ancestral) sequence $x$ (grey, from (A) above) to provide additional upstream/downstream search space for the ②→3 fragment. Template switch alignment then follows Algorithm B in S1 Algorithms. For clarity, initialisations and recursive calculations are only illustrated for the match ($M$) matrices. Note the reverse complement alignment (top right to bottom left) in $M_2$. The unidirectional and TSA pairHMM alignments for this event are given under Event 124 in S1 Data.
(TIF)

**S1 Table. Proportions of gold standard template switch events corresponding to different event types.** Event types are defined by ancestral switch point ordering, and the ensuing rearrangement patterns observed in the descendant sequences. Some pairs of event types are indistinguishable without knowledge of the direction of replication during which an event arose. We indicate these 'mirror cases' as pairs in parentheses. Events that can arise through intrastrand switching are indicated by a preceding *. See [15] for further details.
(PDF)

**S2 Table. Details of human-specific genomic features used for enrichment analysis.**
(PDF)

**S1 Algorithms. A description of the procedure used to define sequence regions for realignment under each model, followed by details of the (A) Viterbi and (B) Viterbi-like algorithms used for the unidirectional pairHMM and the TSA pairHMM, respectively.**
(PDF)

**S1 Data. Unidirectional and TSA pairHMM alignments for all significant events.**
(TXT)

**S2 Data. Human genome (GRCh38.p12) coordinates of the mutation clusters associated with each significant event, in BED format.**
(TSV)

**S3 Data. BED formatted sheets for each of the genomic features processed as described in S2 Table.** In each genomic feature sheet, we report any intersections between that feature and any significant event-associated mutation cluster coordinates in the GRCh38.p12 genome.
(XLSX)

## Acknowledgments

We thank Ari Löytynoja for advice during this study.

## Author Contributions

**Conceptualization:** Conor R. Walker, Nick Goldman.

**Formal analysis:** Conor R. Walker.

**Funding acquisition:** Nick Goldman.

**Investigation:** Conor R. Walker.

**Methodology:** Conor R. Walker, Aylwyn Scally, Nicola De Maio, Nick Goldman.

**Project administration:** Nick Goldman.

**Software:** Conor R. Walker.

**Supervision:** Aylwyn Scally, Nicola De Maio, Nick Goldman.

**Writing – original draft:** Conor R. Walker.

**Writing – review & editing:** Aylwyn Scally, Nicola De Maio, Nick Goldman.

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
