## [Decision Letter · Decision Letter 0]

6 Jan 2021

Dear Dr Goldman,

Thank you very much for submitting your Research Article entitled 'Short-range template switching in great ape genomes explored using a pair hidden Markov model' to PLOS Genetics.

The manuscript was fully evaluated at the editorial level and by three independent peer reviewers. The reviewers appreciated the attention to an important topic but identified some concerns that we ask you address in a revised manuscript.

We therefore ask you to modify the manuscript according to the review recommendations. Your revisions should address the specific points made by each reviewer.

[LINK]

Yours sincerely,

Jianzhi Zhang

Associate Editor

PLOS Genetics

Bret Payseur

Section Editor: Evolution

PLOS Genetics

Reviewer's Responses to Questions

**Comments to the Authors:**

Reviewer #1: This manuscript is a pleasure to read. It is clearly written and reports a valuable development - using a probabilistic model, the authors substantially improve detection of template-switch events which masquerade as clusters of small-scale mutations. Should be published.

My only suggestion is that the authors might want to search for footprints of template-switches in within-human polymorphisms. I understand very well that even ~1% differenes between humans and other great apes make detection of such switches in the course of their diveregence problemattc - we are encountering the same problem looking for microinversions on the human-chimpanzee path. Here, data on human polymorphisms can be really helpful - because with the decline of distance between the compared sequences the "noise problem" must diminish faster than linearly, because it takes multiple independent small-scale events to imitate a template switch. Difference between two human genotypes is ~0.1%, so, extrapolating from what the authors report, I must be heterozygous by dozens of template-switch mutations, which should be much easier to detect. Also, it might be interesting to screen complex de novo pathogenic events (Hum Mutat. 2009 Oct;30(10):1435-48) for template switches.

Alexey Kondrashov (I sign all my reviews)

p. 3 typo - "(25). the occurrence of a single template switch event is".

p. 18 line 6th from bottom - "the find".

Reviewer #2: Review of “Short-range template switching in great ape genomes explored using a pair hidden Markov model”

In this manuscript, the authors develop a probabilistic model that can be used to identify clustered mutations caused by short-range template switching. This model builds upon previous work by the senior author, allowing for more accurate inferences in real datasets.

Overall, I really enjoyed this study. The work is generally well explained and well done, and the results are not oversold or overinterpreted. Despite this positive view, I had a few concerns that I would like to see addressed:

-The model selection procedure used here compares a "unidirectional" pairHMM with the template switch alignment ("TSA") pairHMM. The unidirectional pairHMM represents a kind of null model that only includes independent substitutions and indels, such that model selection is a comparison of the relative probabilities of the data under the two models. However, as mentioned in the Introduction, many clustered mutations are likely caused by error-prone polymerases, which cause both substitutions and indels.

So my concern here is that both the simulations used to choose thresholds are inaccurate, and that inferences from data are also inaccurate. Is it possible that the TSA pairHMM is preferred when in fact the mutations are caused by error-prone polymerases?

While I understand that the TSA pairHMM has to find a likely template nearby, there are no details given about how close this template must be (or how long the alignments used are). And since most clustered mutations are close together, it seems possible to find highly similar templates given enough search space. I also thought that the conclusions could have brought clustered mutations caused by error-prone polymerases back into focus, rather than implying that all clustered mutations were caused by template switching. Even if this is simply to contrast the smaller distances over which error-prone polymerases are likely to act relative to template switching, giving the reader a broader view would have helped.

I would like to see these issues addressed, as well as additional details about what "short range" really means for this method.

-This is a more minor issues, but: should we worry about the mosaic nature of genome assemblies? In other words, Clint (the chimpanzee used for the reference genome) is a diploid, but the assembly is represented as haploid. Should this fact make us more wary of inferences about intra- and inter-strand templating? I'm not clear on all the mechanisms, so wasn't sure if this is an issue.

-The manuscript seems to report that ~94% of templated mutation clusters introduce either an insertion or deletion. Given this, I wondered how often this type of mutation would really be mistaken for sites of accelerated evolution--mostly alignments with indels are discarded from such scans. I would ask that the authors address this in the Conclusions.

-Given the subject matter of the manuscript, I found it surprising that references to Schrider et al. (2011; Current Biology), McDonald et al. (2011; PLoS Biology), Harris and Nielsen (2014; Genome Research), and Besenbacher et al. (2016; PLoS Genetics) were not included.

Reviewer #3: This highly interesting paper adds rigor to the approach of Löytynoja & Goldman (2017) in identifying local mutation clusters that are plausibly explained by template-switching events. A template-switching event is a local indel (or cluster of substitutions) that is actually a nonlocal* inverted duplication, caused by an Okazaki fragment getting grabbed from some other part of the genome instead of the upcoming stretch of the lagging strand during DNA replication. I put an asterisk beside "nonlocal" because, although the duplicated sequence is typically nonlocal in the linear sequence, there is some justification for expecting it to be proximal in 3D space, given knowledge of the 3D structure of the chromosomes. There is in fact extensive study of the various structural mechanisms that can be involved in such events and the complex mutation patterns that can arise, which the present paper briefly reviews in the introduction, but the work itself uses a simpler model (following Löytynoja & Goldman 2017) which just allows the alternate template sequence to be copied from some finite-width neighborhood around the location of the event. This seems to be a reasonably compromise to using actual 3D proximity, and captures a lot of events.

The basic outline of the paper is to present the model, assess its statistical power by simulation, apply it to identifying template-switching events in great ape genomic alignments, and discuss the phylogenetic placement of these events (and the related issue of reversibility). Compared to Löytynoja & Goldman (2017), the new probabilistic method is shown to detect more events and also to have better phylogenetic resolution of when the events occurred. The authors then investigate the properties of the DNA surrounding the "gold standard" subset of imputed template switch events that were unambiguous. This allows the authors to make data-justified claims about the structural properties of DNA around such events. The authors also analyze the proximity of their imputed template-switching events to regions of interest in human genome evolution, such as the "human accelerated regions" and "primate accelerated regions" identified in other studies, and report statistical associations with certain categories of annotated sequence features; perhaps unsurprisingly, this analysis seems to indicate a selection against such events within exons.

All in all, this is very interesting work that makes a significant contribution to our collective understanding of the evolution of genomes, and the human genome in particular. The paper is extremely clearly written, and the scientific rationale is well-referenced and precise. I noticed only one typo (on page 3, the sentence beginning "the occurrence of a single template switch event..." should be sentence-cased, i.e. the "t" should be uppercase).

The critiques/comments/suggestions I have are mild, and mostly relate to the methodological exposition. Perhaps the most significant of these is that I do not believe the authors' algorithm for scoring template switch events (Supplementary Algorithm 2) is a Viterbi algorithm for a Pair HMM, as they claim. Rather it is a compilation of three recursions, each of which could be represented as a Pair HMM. The fact that the model switches to reverse-complement mode halfway through means that it can't be an HMM. However, taken together, I am pretty sure that these three recursions *can* be represented as a Pair Stochastic Context-Free Grammar (Pair SCFG) whose alignment path has been constrained to fit the supplied alignment. Such alignment-constrained Pair SCFGs have been used for other purposes; for example, noncoding RNA gene detection (Rivas & Eddy, BMC Bioinformatics, 2001; Holmes, BMC Bioinformatics, 2005; Dowell & Eddy, BMC Bioinformatics, 2006). A Pair SCFG can readily model a region of a pairwise alignment where the ancestral sequence has been inverted (or partially duplicated via an inverted duplication, as in this case) so this would be the natural way to cast the model, I think.

Why does this matter? Well, one assumes that the point of using standard terminology is to foster continuity with literature & thus maximize the accessibility and impact of the work. Given that Pair HMMs are very standard models, a natural way for a methods-inclined bioinformatics reader to approach this work would be to latch onto the description of the model and attempt to understand it through that lens. In fact, this is how I approached the paper, and found myself wondering "but how does the algorithm decide where the alternate template comes from? A Pair HMM can't do that". It turns out that the answer is that the algorithm actually involves three different recursions that are sort of being painted onto a Pair HMM, and this info is buried in the Supplement. By representing this as a Pair SCFG, the probabilistic formulation would be more rigorous. If the authors find that the Pair SCFG representation is not exactly identical to what they've done, but is close, then I think it'd be acceptable to say that their algorithm is an approximation to a Pair SCFG or is "Pair HMM-like". However misuse of statistically precise terms is somewhat endemic in our field and so I do think that the authors need to qualify that what they're calling a Pair HMM is not the standard Pair HMM that readers may be familiar with (from, e.g., Durbin et al's 1998 book).

(It also seems possible that representing the model more rigorously as a Pair SCFG might lead to a probabilistic formulation of the interesting results in Supplementary Table 1 concerning the relative frequency of different kinds of template-switching events, and co-location of clusters of such events. The proportions in the final column of Supplementary Table 1 could, I imagine, be readily translated to probabilities in Pair SCFGs. However that is probably beyond the scope of this paper.)

Along these lines, thinking about probabilistic modeling rigour, it seems odd to me that - even though the point of this method is to develop a statistical test based on a Bayesian model comparison - the authors' algorithm does not sum over all possible locations of the template switch event, but rather uses a Viterbi algorithm to pick the single most likely location. Using the Forward algorithm instead (or, more precisely, the Inside algorithm for a Pair SCFG) would give a clearer picture and might even boost the statistical signal for "there is a template switch event here" even in cases where there is insufficient statistical power to resolve the precise location of the alternate template (e.g. for low-complexity sequence, or cases where there've been a lot of substitutions within the template switch event, or cases where there are tandem repeats so there are multiple places the sequence could have come from). This is a nuance, and certainly doesn't invalidate what the authors of done; indeed a lot of their subsequent analysis involves is restricted to "gold standard" imputations where the location of the template switch is unambiguous, for which Viterbi-style maximum-likelihood inference is clearly sufficient. So, a minor detail perhaps, but I think worth noting.

With regards to the simulation-based study of how well the method performs, the text mentions that "many template switch events are obfuscated by surrounding neutral mutations, allowing us to capture an average of 78%" of events. This is reminiscent of the "gap wander" phenomenon in pairwise alignment, whereby substitution events that occur near gaps can lead alignment algorithms to misplace the gaps; and if insertions and deletions occur near enough to one another, they can "cancel each other out" and the gap can be missed entirely. This leads to a statistical upper bound on the accuracy of sequence alignment. These phenomena can be analyzed quite rigorously, and in fact Gerton Lunter has derived a closed-form expression for gap wander under the Jukes-Cantor substitution model (Lunter, Rocco et al; Genome Research, 2008) including an analysis of the expected accuracy of human-mouse alignments. This makes me wonder whether these results could be useful in deriving some analytic bounds on the present method. For example, if the 78% figure that the authors report from their simulations could be related to a closed-form expression, it would be a slightly more useful result. I ought to emphasize that this is an open-ended mathematical challenge with no guarantees of a useful result, so I would not insist on it as a revision.

To summarize: I find this a fascinating paper in evolutionary genomics, and a useful methodological advance that significantly improves on the previous work in this area by Löytynoja & Goldman. The results on DNA structure surrounding duplications, and associations with human sequence annotations, are interesting biologically. The work is generally presented very clearly, though I recommend that the authors distinguish Figure 2b from a true Pair HMM and perhaps (assuming they agree with my assessment of this) they might note that it is technically more similar to an alignment-constrained Pair SCFG. I would also recommend that they make some remark explaining why they chose Viterbi/CYK instead of Forward/Inside, and whether that is expected to affect the results. Finally, it might be interesting to try and understand their simulation results using Lunter's analytic theory of gap wander, though this may be a stretch and I wouldn't insist on it.

Ian Holmes

**Have all data underlying the figures and results presented in the manuscript been provided?**

Reviewer #1: Yes

Reviewer #2: Yes

Reviewer #3: Yes

PLOS authors have the option to publish the peer review history of their article (what does this mean?). If published, this will include your full peer review and any attached files.

Reviewer #1: **Yes: **Alexey Kondrashov

Reviewer #2: No

Reviewer #3: **Yes: **Ian Holmes

---

## [Editor Report · Decision Letter 1]

10 Feb 2021

Dear Dr Goldman,

We are pleased to inform you that your manuscript entitled "Short-range template switching in great ape genomes explored using pair hidden Markov models" has been editorially accepted for publication in PLOS Genetics. Congratulations!

Yours sincerely,

Jianzhi Zhang

Associate Editor

PLOS Genetics

Bret Payseur

Section Editor: Evolution

PLOS Genetics

Comments from the reviewers (if applicable):

**Data Deposition**

http://datadryad.org/submit?journalID=pgenetics&manu=PGENETICS-D-20-01724R1

**Press Queries**

---

## [Editor Report · Acceptance letter]

23 Feb 2021

PGENETICS-D-20-01724R1 

Short-range template switching in great ape genomes explored using pair hidden Markov models 

Dear Dr Goldman, 

We are pleased to inform you that your manuscript entitled "Short-range template switching in great ape genomes explored using pair hidden Markov models" has been formally accepted for publication in PLOS Genetics! Your manuscript is now with our production department and you will be notified of the publication date in due course.

With kind regards,

Alice Ellingham

PLOS Genetics

On behalf of:
